# Where is emotional feeling felt in the body? An integrative review

**Steven Davey** [1]*, **Jamin Halberstadt**[2], **Elliot Bell**[3]

**1** Department of Psychological Medicine, University of Otago Wellington, Wellington, New Zealand,
**2** Department of Psychology, University of Otago, Dunedin, New Zealand, **3** Department of Psychological
Medicine, University of Otago, Wellington, New Zealand

* daveystev@myvuw.ac.nz

Where is emotional feeling felt in the body? An
integrative review. PLoS ONE 16(12): e0261685.

UNITED STATES

**Data Availability Statement:** All relevant data are
within the paper and its Supporting Information
files.

**Funding:** The authors received no specific funding
for this work.

## Abstract

Contemporary research on "embodied emotion" emphasizes the role of the body in emotional feeling. The evidence base on interoception, arguably the most prominent strand of embodied emotion research, places emphasis on the cardiac, respiratory and gastrointestinal systems. In turn, interoception has evidence-based links with improved emotion regulation. Despite the focus on separate bodily systems, it is unclear whether particular interoceptive locations play a greater role in emotional feeling and emotion regulation. Further, according to Gross' "process model", the sooner that regulation of an emotion occurs, the better; hence, it is additionally important to identify the first body areas to activate. These issues are investigated in a two-stage integrative review. The first stage was preliminary, giving an overview of the evidence base to highlight the distribution of measured body areas. This indicated that 86% of publications ($n = 88$) measured cardiac activity, 26% measured the respiratory system, and six percent the gastrointestinal system. Given the emphasis placed on all three systems in interoception theory and research on emotion, this suggests a dearth of comprehensive findings pertaining to feeling locations. The second stage investigated the core issues of where emotional feelings are felt in the body and time-related implications for regulation. This was based on ten texts, which together suggested that the head, throat and chest are the most consistently detected locations across and within numerous emotional contexts. Caution is required, however, since–among other reasons discussed–measurement was not time-restricted in these latter publications, and direct physiological measurement was found in only a minority of cases.

## Introduction

With the exception of a period dominated by behaviorist reductionism, subjective emotional experience has been a focal point in psychology since its inception as a separate discipline (with Wundt's studies on introspection). Emotional experiences are routinely captured using self-report tools and objective measures such as cognitive tasks, behavioral observation and physiological measurement. As an advance on the introspective approach to examining feelings, William James emphasized the intersection between subjective emotional feeling and the

**Competing interests:** The authors have declared that no competing interests exist.

objectively measurable features of emotion, leading to what is now referred to as "embodied emotion".

According to James, emotion is the feeling of bodily changes that arise immediately from "perception of the exciting fact" [1]. For example, there is the distinction between *running* from a bear because we are *afraid*, and being *afraid* because we *run*. Running is the response to the "exciting fact" (i.e., a bear) and feeling the overall embodied response *is* the emotion. According to James, then, emotion is not a separate state that causes bodily responses; rather, emotion consists of bodily responses. Modern scientists such as Damasio [2] have revitalized and extended James' perspective, with articulation of an account involving afferent feedback from bodily associations, which has resulted in the significance of the body in the contemporary psychological study of emotion. In particular, Damasio's "somatic marker hypothesis" suggests that some bodily responses are the translation of emotional information from the autonomic nervous system to conscious awareness, acting as "biasing" bodily signals to mark response options that require full processing [2]. Somatic markers boost attention, thereby amplifying environmental features (including James' bear). Hence, such markers act as an interface between subjective experiences and objective events.

One prominent construct for understanding the interface between subjectivity and objectivity, and thereby consistent with the embodied view of emotion, is *interoception*. This has been defined as "the sense of the physiological condition of the body" [3], which makes a vital contribution to emotional experience [4]. Arguably the most prominent articulation of interoception in the literature is that provided by Garfinkel and colleagues [5]. This articulation is also clearly consistent with the embodied emotion view of James and Damasio, with an emphasis in all cases on objective bodily changes providing the foundation for subjective emotional experience. According to Garfinkel and colleagues, interoception is decomposable into three aspects [5]:

- interoceptive sensitivity (the ability to accurately detect physiological changes, using objective detection measures),

- interoceptive sensibility (the self-reported subjective sense of physiological changes, measurable by tools such as sub-scales of the *Multidimensional Assessment of Interoceptive Awareness*, or MAIA; [6]),

- metacognitive awareness (the degree of correspondence between measurements of sensitivity and sensibility).

Originally, "interoception" pertained to only the viscera [7] whereas it is now understood to pertain to locations throughout the body, but especially the viscera [8]: the "visceroceptive" processes of "cardioception" (interoception of the heart), respiratory interoception (interoception of respiration rate, inspiratory resistance etc., regarding the activity of the lungs and surrounding structures) and "gastroception" (interoception of the gastrointestinal tract). In relation to research on interoception and emotion, specifically, the three visceroceptive domains (heart, lungs, gut) have to date been prominent focal points, including within clinical studies such as those involving the viscera and panic attacks (see [9]). For example, in an overview of the literature on emotion and interoception, Critchley and Garfinkel [10] emphasised the three visceral systems, and particularly the dominance of the heart in interoceptive research. The latter is likely due in part to the availability of non-invasive heart measurement (e.g., electrocardiogram, or ECG, and self-report). There is, for example, the "Schandry task", which involves counting one's heartbeats within varied time frames for comparison with objective heartbeat measurements [11]. Greater cardioceptive ability is measurable using this task, where such ability has been linked to subjective emotional experiences, such as increased

emotional intensity and arousal [12,13]. The Schandry task is relatively easy to administer and score, thereby likely compounding cardiac dominance in interoception-emotion. Whilst emphasis on the cardiac system is unsurprising, nor by extension is an emphasis on viscero-ception per se surprising, given the physiological relationships between the cardiac, respiratory and gastrointestinal systems, including common vagal connections.

Measuring interoception of non-cardiac visceral locations and systems has, however, pre-sented greater difficulty, with these having received relatively less attention than the heart. Despite the invasiveness of, for example, inspiratory load, some research has nevertheless focussed on the respiratory system, such as the use of breathing practices in meditation, and their emotional effects (e.g., [14,15]). Notably, Daubenmier, Sze, Kerr, Kemeny, and Mehling [14] developed an interoceptive procedure (a "respiratory tracking task") that does not depend on applying a respiratory load, and is no more invasive than ECG. Other work has focussed specifically on respiration in clinical conditions, such as panic disorder, including research examining the fear of dyspnoea ("air hunger"), potentially precipitating panic attack in people with chronic obstructive pulmonary disease (COPD) [16].

Similarly, the gastrointestinal tract has been studied to a limited extent, primarily in the context of clinical conditions (e.g., irritable bowel syndrome; [17]), rather than its role in inter-oception-based emotional feeling, per se. However, there are studies showing a link between the gastrointestinal tract and emotional experience (e.g., [18])–again unsurprising given the vagal innervation of the gut, plus the emerging evidence on the emotional role of the "enteric nervous system" (e.g., [19]), and the gut microbiome [20]. Finally, as with the heart and respi-ratory systems, there are non-invasive ways to measure gastrointestinal activity, principally the electrogastrograph (EGG), which measures gastric myoelectrical activity. Hence, despite a rela-tive paucity of gastroception research, opportunities exist to conduct such research without thereby creating additional procedural complexity.

Regarding the aforementioned visceral systems, the evidence base is equivocal regarding their interrelationship. Whilst correlations have, for instance, been found between cardiocep-tive and gastroceptive sensitivity [21–23], and between the cardiac and respiratory systems regarding metacognitive awareness [24], there is much counterevidence. Indeed, objective measures involving these channels often do not result in correlation, particularly regarding interoceptive sensitivity (the most frequently studied aspect of interoception); as Gibson points out in his review, the evidence on this issue is both heterogeneous and limited [25]. For instance, the cardiac and respiratory systems have been found to be only weakly correlated in terms of interoceptive sensitivity [24]. Garfinkel and colleagues [26] found an association between cardiac and gastrointestinal sensitivity, but no association between these two channels and respiratory sensitivity. Also, Ferentzi, Bogdány, Szabolcs, Csala, Horváth, and Köteles [27] found no between-channel associations at all–across multiple channels–including cardiac and gastrointestinal. Such studies challenge the view that interoceptive sensitivity is a general abil-ity; rather, interoceptive ability may be channel-specific. A corollary of this is that high levels of cardioception need not translate to generally high levels of interoceptive sensitivity (across any/all channels); hence, there is a need to investigate multiple channels in studies of intero-ception, thereby challenging cardiac dominance.

Given its links with emotional experience, interoception has, in turn, been associated with emotion regulation. Emotion regulation is a process of modulating aspects of emotion (i.e., thoughts, feelings, behaviours) to achieve situational adaptation [28]. Lower levels of intero-ceptive sensibility have been associated with a reduced ability to regulate emotion [29]. Higher levels of interoceptive sensitivity have been associated with greater emotion regulation [30] and linked [31] to two components of the "process model" of emotion regulation [32]. Accord-ing to this, the regulation process involves a distinction between "antecedent-focussed"

strategies and "response-focused" strategies [33]. Antecedent-focussed regulation is an early four-stage modulation sequence, prior to the embodied emotional case of "multi-componential syndromes" where numerous body parts and systems become active, and with response-focussed regulation occurring after a response is underway. In terms of cognitive strategies, at least, the antecedent-focussed approach appears to have the best outcomes [33], which is likely due to greater efficiency, given that such strategies are implemented at a point prior to the need for responding to a full-blown emotional episode requiring greater cognitive resourcing [34,35]. One such strategy involves the attentional deployment stage of the "process model": either attending to or redirecting one's attention from the early genesis of an emotion/situation. In terms of interoception and its role in emotion regulation, then, it may be that an antecedent approach (e.g., a training in interoceptive awareness), has the best chance of success, with early stage conscious awareness of physiological change during an emotional response being preferable to a later stage escalation of bodily activation, in order for there to be successful adaptation.

Further, there may be clinical applications regarding interoceptive contributions to emotion regulation. Such applications may involve bringing underlying emotionally relevant signals into awareness for conscious processing [36]. There is, for instance, growing evidence that emotion dysregulation is a significant factor in suicidality and, more generally, on the negative consequences of emotional disconnection in relation to interoceptive deficits in people experiencing suicidality. Conversely, the benefits of *reconnection* with embodied emotion are also indicated in relation to suicide prevention [37]. There are numerous studies showing the psychological and clinical value of interoception, such as when combining interoception and dispositional mindfulness (e.g., [38]), or with Mindful Awareness in Body-oriented Therapy (MABT) (e.g., [39]), involving techniques such as the "body scan" (a sequential focus on body areas), which has been shown to improve interoception [40]. Some studies have, however, suggested that focusing on the body may not always be positive, such as when detection of sensations by interoception exacerbates social anxiety [41]. However, this may be due more to pre-existing cognitive influences pertaining to worry about the body, rather than a problem of excessive interoception (see, for instance, [42]).

One challenge for any form of emotion regulation is that emotional responses develop rapidly, even those involving potentially sensed activation of the body. The timeframes within which responses unfold are evident in studies such as Codispoti, Mazzetti & Bradley [43], in which an unmasked emotional stimulus (from the *International Affective Picture System*, IAPS; [44]), produced reliable arousal levels (assessed using self-report and physiological measures) even at 25ms presentation. Even in a masked condition, similarly early signs of activation resulted after an 80ms presentation time [43]. Further, in the unmasked condition, arousal did not increase significantly beyond 150ms post-onset; subjective ratings were provided within a limited window of 10-second post-stimulus onset. Hence, subjective ratings reflected underlying physiological changes, all undertaken within tight timescales, providing evidence of early stage sensing of emotional changes in the body. Relatedly, another study using the IAPS [45] indicated the mean arousal rating required to elicit emotionally relevant physiological changes, in this case heart rate variability. A score of $> = 4.5$ was required, out of a maximum of 5 on the Self-Assessment Mannikin (SAM; [46]). Such research indicates, then, the possibility of eliciting genuinely embodied emotional experiences using standardized tools, experiences that appear possible during the earliest stages of emotional responses. The awareness of these early stage changes could provide the basis for regulation of an otherwise potentially rapid non-adaptive escalation of emotional responding.

Overall, then, the links between subjective emotional feeling, emotion regulation and the body have been well explored in recent years, which includes the rising importance of

interoception, most prominently articulated in the tripartite form developed by Garfinkel and colleagues [5], as well as other forms such as the multidimensional account of Mehling and colleagues [6]. Independently of any one articulation are the differing forms of interoceptive practice; i.e., different focal points in and across the body. Rather than the "body scan" of mindfulness or some other generic body focus, there may be additional regulatory value in focusing on a specific part or parts of the body. To our knowledge, however, whilst interoceptive research implies the importance of bodily location with the emphasis on the heart, the gut, and lungs, whether these or other locations/bodily systems play distinctive roles in emotional experience has yet to be explored. Further, whilst there is a general sense that the heart dominates interoception research–and perhaps embodied emotion research more generally–the extent to which this occurs is unclear. With the evidence base pointing to channel-specific interoceptive ability, there is little sense in limiting focus to the cardiac system. Another implication of cardiac dominance is that with current measures such as the Schandry task having been questioned regarding their reliability (e.g., [47]) cardioceptive dominance may severely inhibit progress in interoception research. Finally, it is unclear which–if any–bodily location merits more attention regarding activation in emotional experience and how location may in turn relate to optimal emotion regulation.

## Aims

The primary focus of this review is the relationship between emotional feelings and body locations, including the heart, lungs and gut, which have to date been prominent focal points in the field of interoception. The review seeks to identify publications addressing body locations including, but not limited to, visceral locations, the awareness of which could play a role in effective emotion regulation, especially as an early-stage response strategy. A prima facie "gold standard" publication would be one where at least a majority of overall body locations implicated in the literature (including all three visceroceptive systems) are measured within tight time-scales following the presentation of emotional stimuli, with accompanying self-report measurement for those same body areas to establish corresponding subjective sensations, thereby grounding sensation in objective activity. There may also be grounds for including self-report only studies with innovative designs that are able to capture subjective reports of genuine bodily sensations even without physiological measurement. These may, for instance, draw on the findings cited previously regarding brief stimuli presentation timescales and required arousal levels for physiological activation. In this way, whilst the current review emphasises the tripartite model of Garfinkel and colleagues [5], as a prominent viewpoint, no absolute claims are asserted about its accuracy. This is particularly important, given that the construct of interoception remains under development (see, e.g., [48]). Such development is indeed required, given findings on a lack of correspondence between interoceptive sensitivity and sensibility; i.e., high scores in the former may not yield high scores in the latter [5]. Indeed, Murphy, Catmur, and Bird [48] suggest that the lack of correspondence may relate to the distinction between attention and accuracy; in other words, it matters as much *what* is being measured in addition to how. Outside of these debates, the review is committed to the view that subjective experiences of emotion are grounded in bodily changes–this is a more fundamental commitment, grounded in the original view of James [1] and that of Damasio [2].

Secondly, the review seeks to provide an initial overview of which body locations are routinely measured in emotion research, in cases where subjective feelings of those locations could occur, at least in principle. This is to set the scene for the primary aim by identifying the extent to which the literature is dominated by certain body locations [10] to the exclusion of others, with the potential for biasing embodied emotion research via a self-reinforcing process

of limiting measurement to those locations (such as when driven by attempts to replicate). At the same time, this can contextualize the situation for the three visceroceptive systems that appear to make important theoretical and applied somatic contributions to emotion, specifically. Whilst not a comprehensive treatment of "where emotional feeling is", measuring all three systems in a study would at least provide a reasonably well-rounded assessment of embodied emotion, as a reflection of the current evidence base on interoception-emotion. Since all three systems can be measured non-invasively and with no *a priori* obstacle to concurrent measurement it seems likely such studies have been undertaken, but the extent to which this has happened is an open question.

Against the backdrop provided by this secondary aim, the key research questions of this review are: (i) Where is emotional feeling reported to be located in the body? (ii) In particular, which part(s) of the body is/are active in the earliest stages of emotional feeling?

## Methods

Given the wide-ranging nature of the subject matter, with emotional responses occurring across several levels–i.e., subjective sensation, behavioural responding, physiological responses–this review includes a relatively wide range of subject areas. However, with the focus being bodily *sensations* (i.e., subjective feelings), purely neuroscientific studies are not in scope, although physiological studies, more generally, were included provided they pertained to potentially sensed body locations. In addition to relevance criteria, information relevant to the quality of research reports was extracted but not used to exclude studies. Given the diversity of publications in scope, only general standards for research quality were applicable (i.e., assessments of sample size, clarity of aims, clarity of outcome etc. Quality assessment was based on Jadad and colleagues [49]). With a requirement that studies were peer-reviewed publications, pre-existing quality assurance was, to some extent, already in place prior to assessment of individual texts.

The general prerequisites for inclusion were then as follows:

- The subject area was limited to psychology/psychiatry; medicine/physiology.

- The selected studies were empirical (i.e., not philosophical, conceptual, theoretical or symbolic), including systematic reviews of empirical investigations.

- Embodied emotion was investigated (given the empirical requirement, this excluded an undefined role for the body, or meaning/symbol it represented).

- An explicit emotional stimulus was used to elicit emotional responses in a (potentially felt) bodily location as a measured outcome.

- Emotional stimulus was non-tactile, thereby not involving direct manipulation of the body area to be measured (i.e., to avoid measurement artefacts, in the context of current aims), such as cases of administering a drug causing tachycardia with heart activity as an outcome measure).

- Human studies.

- English language, peer-reviewed publications.

The review channels publications into two main groups. There is a first group of "general" texts, which had to meet the above criteria. These are intended to permit assessment of the extent to which certain body areas dominate embodied emotion research, and where others may be under-represented. Beyond this overview of the literature, there is then a second separate main group of "core" texts, which specifically focus on the key question of this review, of

where emotional feelings/sensations manifest in the body, as an outcome measure, in addition to meeting the general relevance criteria. To be considered comprehensive, a set of essential locations are as follows: (i) heart/cardiac; (ii) lung/respiratory; (iii) gut/gastrointestinal; (iv) skin (i.e., due to widespread use of skin conductance and galvanic skin response in emotion research); (v) eye/face (i.e., widespread use of electromyography, startle/blink response, facial expressions), and (vi) head (i.e., as reported in a relevant study, [50], and due to the current authors' own data from a study related to the current review, during which participants were asked to indicate where they ". . .most tend to detect emotional feeling", with many indicating the head).

The search strategy involved four separate stages: (i) a large scale initial set of searches following collaboration with medical library specialist staff at the University of Otago Wellington (UOW) to develop bespoke sets of search terms for each of the databases accessed (databases accessed 20th February 2020); (ii) a subsequent search of gut/gastrointestinal sources (for reasons stated in the results section; databases accessed 2nd April 2020); (iii) searching reference lists to identify further sources; (iv) a final set of searches based on the measurement tools used in the near-final set of core texts (databases accessed 5th June 2020). The databases accessed were: MEDLINE, PsycInfo, EMBASE, EBSCO (behavioural sciences), SCOPUS, the Cochrane Database of Systematic Reviews, and the Cochrane Central Register of Controlled Trials.

For the first stage of searches, each publication was sifted initially by title, followed by abstract, then by full text, with publications streamed into one of the groups of texts as appropriate. Those passing all three stages went forward to a final integrative/synthesis stage. For this main set of searches, a 10% check was undertaken by a colleague to ensure a consistent and appropriate application of text selection criteria. The second search stage used keywords and titles from the original search/sift, applying these across the same databases as previously, but accessing specifically gut-focussed publications. The third stage accessed abstracts and full texts from the reference lists of all core and publications measuring all three visceroceptive locations, to identify further core texts. After the first reference searches, the process was repeated until no additional core texts were identified. (This resulted in three reference list searches.) The fourth and final stage was based on the set of core texts, which (as stated in the results) formed three clear groupings, arranged around distinct types of self-report measurement: (i) body mapping, (ii) a "Scene Construction Questionnaire", and (iii) a "Visual Analogue Scale" used by authors in combination with a self-report checklist. On that basis, a final set of searches were undertaken, again using all databases from the original searches, involving many of the same keywords and title words used in the additional gut-focussed searches. The combinations of search terms used for each stage are provided in S1 Table.

The resulting core texts were synthesised using the integrative review guidance of Whittemore and Knafl [51], which outlines a five-stage process of: (i) data reduction (i.e., coding extracted information, forming logical sub-groups); (ii) data display (e.g., matrices, graphs; charts); (iii) data comparison (i.e., a process of "constant comparison" to convert extracted data into systematic categories, forming patterns, themes, and variation in relationships); (iv) conclusions (i.e., higher levels of abstraction/generalization); and (v) verification (i.e., confirming conclusions using the primary sources).

For (i) data reduction, subgroups were formed based on a study's methodology (in our case, a questionnaire, a rating scale, or a "body map"). For (ii) data display, matrices were used in Excel with extracted data in separate columns, using tables to summarise extracted data. Texts then underwent (iii) the data comparison process of 'constant comparison' involving memo-writing within a separate column to combine and summarise data from remaining columns for each publication, identifying patterns/variation, before writing between publications to establish links. Using mind-maps, memos were linked to form more general statements for

the collection of publications to arrive at (iv) final conclusions, supported in turn by reference to the primary sources, i.e., (v) verification. Where necessary, the integrative process also involved returning to the original publications to check for accuracy and to extract any additional information to inform each of the five stages.

The final outputs from this process for core texts are a main table displaying a summary of extracted material with an accompanying narrative, plus some additional tables responding to the various findings of a "body map" cluster of publications (for the latter, see S2 Table), and a separate table of those remaining publications using more traditional self-report tools pertaining to bodily sensations (S3 Table).

## Results

The results of the first stage of searching are displayed in Fig 1 (the minimal dataset underlying all results is in S2). From an initial 1777 hits, 51 publications went through to the final list of texts. Following this search it was clear that, in comparison to the heart and respiratory system, far fewer publications than anticipated had undertaken any form of measurement of the gastrointestinal tract or of emotion related symptoms (e.g., stomach ache or nausea). It is for this reason that the second set of searches was undertaken to check whether the original search terms had been adequate for capturing gut-related studies. Together with the two further stages of searching, 47 further texts were added to the original 51 (see Fig 2), resulting in a total number of 98 texts (general and core). The majority of texts met a general quality standard (i.e., clarity of reporting: aims, analyses, experimental design, sampling criteria, findings; having undertaken a power calculation), although relatively few reported on sample size/ power calculations.

### General texts

Of these 98 texts, 88 were classified as general texts (Table 1). A wide array of stimulus types were used across these general publications, including images (e.g., from established emotional stimulus sets, such as the IAPS); sounds (e.g., music, heartbeat); computer games and virtual reality (VR); social scenarios and stressful encounters (e.g., public speaking); films; and the use of startle probes in combination with other stimuli. Measured outcomes were largely as expected, with combined self-report and bodily/physiological locations (Fig 3) showing a predominance of heart/cardiac activity (88% of publications), followed by skin (63%), eye/face/ head combined (40%), and the respiratory system (32%). The respiratory system was rather better represented among studies where outcomes could not be included due to artefacts from tactile stimuli, mainly in studies using respiratory loads and hyperventilation as stimuli in the investigation of anxiety and panic disorder. Despite follow-up searches for the gut/gastrointestinal tract, the final proportion of publications where this was measured was just 15%. The pattern was similar when focussing on only bodily/physiological outcome measurements, but with the gut being yet less represented (falling to 6% of all publications). Overall, three of the general papers reported participants' self-reports on specific body locations as outcomes. These also recorded the corresponding body area using physiological measures, with correspondence identified between self-reported and physiological measures for skin activation [52,53], and heart activation [53,54].

### Core texts

From the full set of searches, 10 publications were classified as core texts (see Table 2). Of these, 7 used "body mapping" tools, mostly the "emBODY" tool originating with Nummenmaa, Glerean, Hari & Hietane [50]. This is a computer-based pixel selection program that

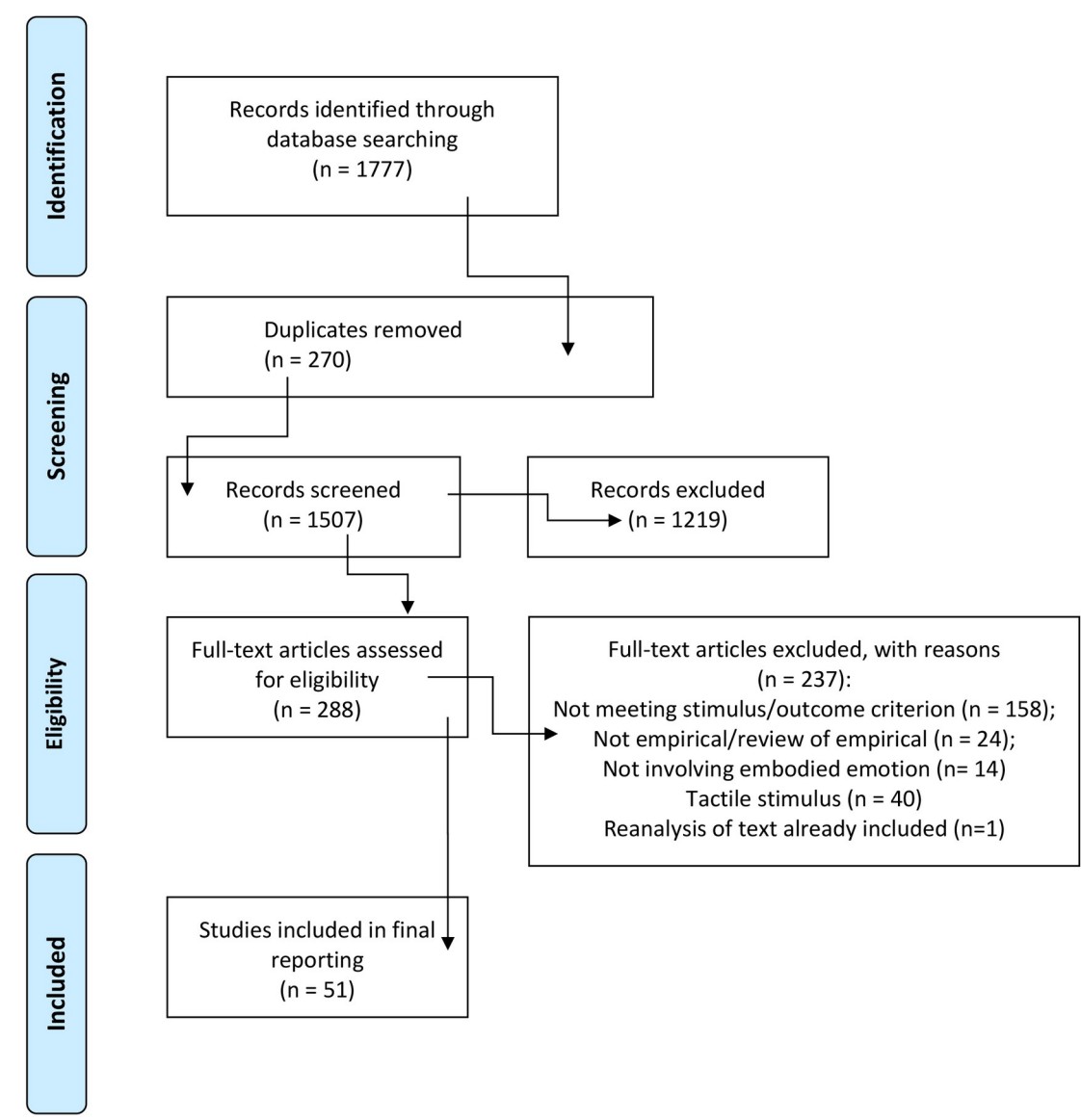

From: Moher D, Liberati A, Tetzlaff J, Altman DG, The PRISMA Group (2009). Preferred *R*eporting *I*tems for *S*ystematic Reviews and *M*eta-*A*nalyses: The PRISMA Statement. PLoS Med 6(7): e1000097. doi:10.1371/journal.pmed1000097

For more information, visit www.prisma-statement.org.

**Fig 1. PRISMA main searches.**

allows participants to register mouse clicks within a body outline (silhouette), one to record a sensation of bodily activation, and one to record deactivation. These silhouettes are combined to create heat maps, with warmer colours indicating increasing activation, cooler colours

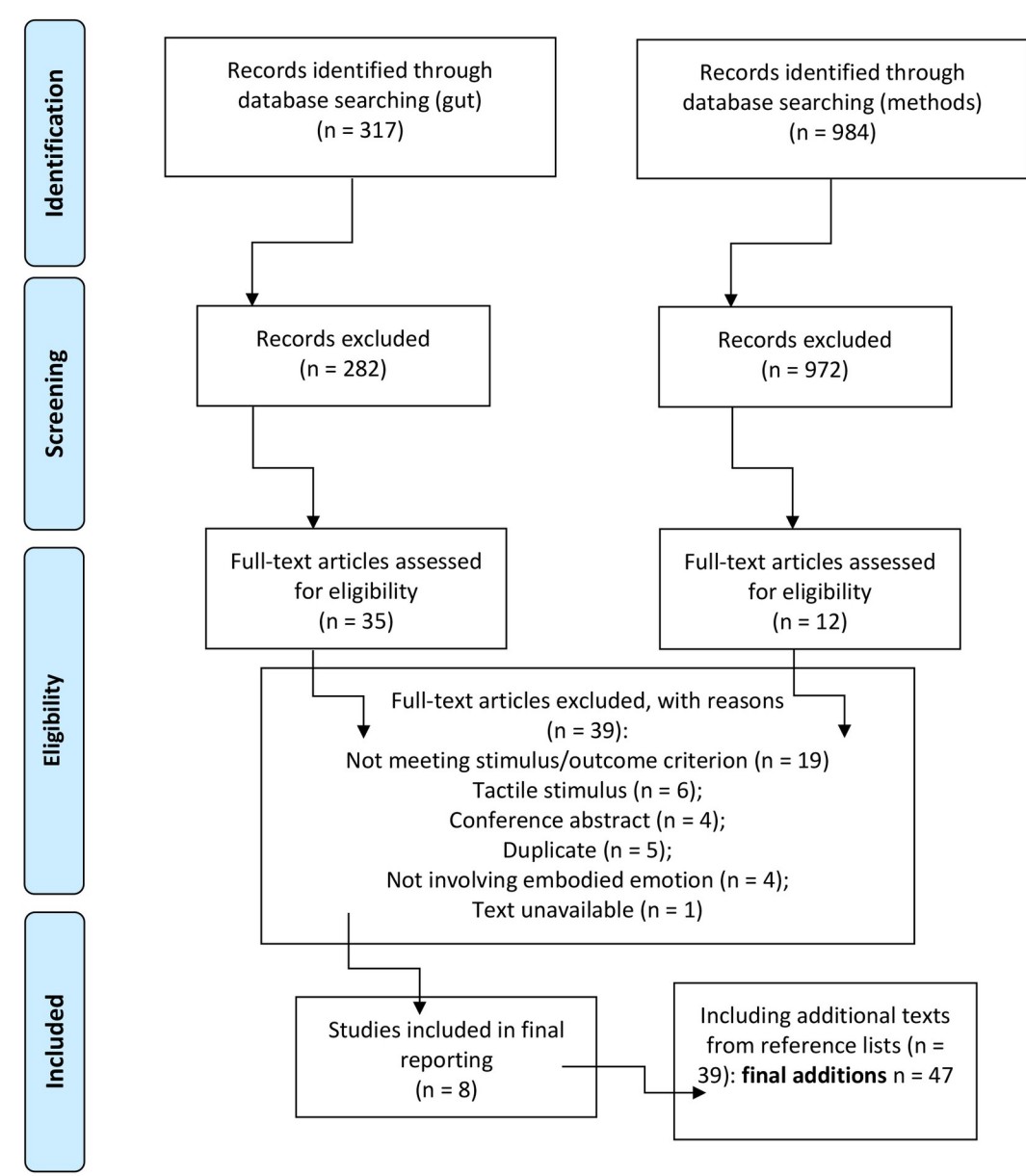

*From:* Moher D, Liberati A, Tetzlaff J, Altman DG, The PRISMA Group (2009). Preferred *R*eporting *I*tems for *S*ystematic Reviews and *M*eta-*A*nalyses: The PRISMA Statement. PLoS Med 6(7): e1000097. doi:10.1371/journal.pmed1000097

**For more information, visit www.prisma-statement.org.**

**Fig 2. PRISMA further searches.**

indicating deactivation, and black indicating neutral. With one exception [139], all body mapping papers used the emBODY software or, in the case of child participants, subsequently transferred a pen and paper version to the program [140,141]. In place of the emBODY tool,

**Table 1. Background texts.**

| Publication | Description of paper | Emotional stimuli | Body location(s) measured | Self-report tool details | Subject area (psych; medic/ physiol) |
|---|---|---|---|---|---|
| Anderson [55] | Investigating startle blink responses (measured by EMG) during affective picture viewing in psychopaths, and ERP | Images | Eye(blink)/face | | Psych/ physiology |
| Baldaro [56] | Used films, measuring cardiac, respiratory sinus arrythmia (RSA), and EGG measurement (electrogastrogram) | Films | Heart; respiration rate; RSA; gut | | Psych/ physiology |
| Bernat [57] | Emotional pictures used, measuring cardiac, respiratory, facial movements, SCR | Images; startle probe | Respiratory; skin; heart; eye (blink)/face | | Psych/ physiology |
| Bradley [58] | IAPS used as emotional stimuli, with startle probe used in half of trials, measuring heart rate, skin conductance, activity over the facial corrugator, zygomatic, and orbicularis oculi muscles | Images; startle probe | Skin; heart; eye(blink)/face | | Psych/ physiology |
| Breuninger [52] | Used VR stress stimulus with HR and SC as physiological responses (after the VR simulation), with some other self-reports on heart intensity and sweating etc. | Virtual Reality (VR) | Skin; heart; self-reports on heart and sweating | Physio measures and self-report of same locations | Psych/ physiology/ medic |
| Chaplin [59] | Used "scripts" using the Scene Construction Questionnaire, which were played in a laboratory setting. One script was "stressful", another related to drug/alcohol addiction and the third was a neutral control. Used the "Behavioural Arousal Scale" (BAS) and self-report to measure outcomes | Stressful "scripts" | Self-report/behavioural observation (using BAS): respiratory, skin, muscles, head, gut, eye(tears) | Measures taken during the session | Psych/medic |
| Chaplin [60] | Used "scripts" using the Scene Construction Questionnaire, which were played to them in a laboratory setting. One script was "stressful", another related to drug/alcohol addiction and the third was a neutral control. Used the "Behavioural Observation Scale" (BOS) (involving muscle twitching, muscle tremor, restlessness, muscle tension, muscle ache, headache, quickened breathing, yawning, talking / facial movements, crying, sweating, and stomach / abdominal changes), blood pressure, heart rate) | Stressful "scripts" | Heart; self-report (BOS): respiratory, skin, eyes/face, gut, head | BOS: immediate measurement by independent rater. N.B: study meets inclusion criterion for body areas recorded, but data are used as a compound measure, with no specific reporting on locations | Psych/physio/ medic |
| Codispoti [61] | Using emotional images with acoustic startle probe, measured eyeblink response (EMG), heart rate and skin conductance | Images; startle probe | Skin; heart; eye(blink)/face | | Psych/ physiology |
| Codispoti [62] | Using pleasant, unpleasant, high arousal films, measured heart rate, respiratory sinus arrhythmia, skin conductance, EMG | Films | Heart; RSA; skin; eye(blink)/ face | | Psych/ physiology |
| Davis [63] | A range of image types (pleasant and unpleasant) used, measuring heart activity, respiratory and GSR | Images | Respiratory; skin; heart | | Psych/ physiology |

*(Continued)*

**Table 1.** (Continued)

| Publication | Description of paper | Emotional stimuli | Body location(s) measured | Self-report tool details | Subject area (psych; medic/ physiol) |
|---|---|---|---|---|---|
| Dimberg [64] | Used facial affect images, measuring skin conductance response (SCR), EMG and HR | Facial images | Skin; heart; eye(blink)/face | | Psych/ physiology |
| Dunn [12] | Used IAPS stimuli, with ratings of arousal and valence. Physiological: heart rate change (deceleration). Schandry task was tested separately | Images | Heart | | Psych/ physiology |
| Durlik [65] | Whether high levels of anticipation regarding a speech delivered by participants would be associated with high IA (measured by Schandry) | Stress task | Heart | | Psych/ physiology |
| Edelmann [53] | During stressful situations, heart rate, skin conductance, and face and neck temperatures were recorded, along with self-report for within 3 categories relevant to social phobia (racing heart, sweaty hands and body heat) | Stress task | Skin; heart; face; self-report: includes hands | Self-report locations and physiological measures of same locations. No immediate responses apparent | Psych/physio/ medic |
| Fairclough [66] | Changing levels of anxiety and measuring heartbeat detection (two-choice Whitehead paradigm) performance | Stress task | Heart | | Psych/ physiology |
| Foerster [67] | Several stressful tasks, measuring cardiovascular activity, SCR, respiratory activity, eyeblink | Stress task | Respiratory; skin; heart; eye (blink)/facial | | Psych/ physiology |
| Frazier [68] | Whether changes in breathing-related heart changes (RSA) are related to emotional changes (arousal, valence) | Films | Heart; RSA; skin | | Psych/ physiology |
| Füstös [69] | Investigated whether interoceptive awareness (measured by Schandry) helps to improve emotion regulation for negative stimuli (IAPS), using self-report/subjective measures (SAM) and physiological measures (EEG, ECG) | Images | Heart | | Psych/ physiology |
| Gomez [70] | Viewing emotional images and measuring aspects of heart activity (HR, BP, stroke volume, cardiac output, and total peripheral resistance) | Images | Heart | | Psych/ physiology |
| Hackford [71] | This research investigated whether an upright walking posture could change the impact of emotional stressors (Trier Social Stress Test) on emotional outcomes, and physiological states (BP, GSR, skin temperature) | Stress task | Skin; heart | | Psych/ physiology |
| Hare [72] | Images shown, measuring cardiac, respiratory, eye movements and SCR | Images | Respiratory; skin; heart; eye (blink)/facial | | Psych/ physiology |
| Hastings [73] | Investigated coherence between subjective and physiological (heart rate) measurements, using emotional films as stimuli and taking ECG recordings | Films | Heart | | Psych/ physiology |

(*Continued*)

**Table 1.** (Continued)

| Publication | Description of paper | Emotional stimuli | Body location(s) measured | Self-report tool details | Subject area (psych; medic/ physiol) |
|---|---|---|---|---|---|
| Hawk [74] | Using emotional sounds (hums, grunts etc.) as stimuli, facial responses of participants were measured | Sounds | Eye(blink)/face | | Psych/ physiology |
| Hilmert [75] | Using stressors (e.g., public speaking) as emotional stimuli, measuring cardiovascular responses | Stress task | Heart | | Psych/ physiology |
| Kim [76] | Using emotional pictures and music to elicit emotional responses in different physiological systems. Electromyography recordings for the zygomaticus major and corrugator supercilii were measured along with heart rate and skin conductance level | Images; music | Eye(blink)/face; skin; heart | | Psych/ physiology |
| Klorman [77] | Used images, measuring heart activity, skin conductance, respiratory changes and eye movements | Images | Respiratory; skin; heart; eye (blink)/facial | | Psych/ physiology |
| Klorman [78] | Emotional pictures used, measuring cardiac, respiratory and SCR | Images | Respiratory; skin; heart | | Psych/ physiology |
| Korb [79] | Facial mimicry following presentation of emotional face stimuli, measuring facial muscle movement | Facial images/film | Face | | Psych/ physiology |
| Krause [80] | Used respiratory restriction and electric shock as emotional stimuli, along with an approach stimulus (i.e., indicating the approaching threat), measuring skin conductance, changes in respiration, startle response (eyeblink) and heart rate deceleration | Resistive respiratory load; electric shock; approach stimulus; startle probe | Respiratory; skin; heart; eye (blink)/facial | | Psych/ physiology |
| Kreibig [81] | Review of ANS activity following emotional stimuli for the cardiac, respiratory systems, and skin conductance | Numerous | Respiratory; skin; heart | | Psych/ physiology |
| Kreibig [82] | Investigated emotional states in relation to HR, BP, skin conductance, respiration changes, finger temperature, blood volume waveform, finding physiological differentiation by emotion type | Films | Respiratory; skin; heart | | Psych/ physiology |
| Kreibig [83] | Used films, measuring cardiac, respiratory and skin responses, plus facial movements | Films; startle probe | Respiratory; skin; heart; eye (blink)/facial | | Psych/ physiology |
| Lang [84] | Whether viewing emotional images produced patterns of physiological activity, recording eyeblink response, heart rate, skin conductance, for each emotion type, valence and arousal | Images | Skin; heart; eye(blink)/face | | Psych/ physiology |
| Lang [85] | Emotional images used, measuring heart, respiratory, SCR and eye movements | Images | Respiratory; skin; heart; eye (blink)/facial | | Psych/ physiology |
| Lang [86] | Stress tasks and images used, measuring heart activity, respiratory activity, SCR, facial movements | Stress task; images | Respiratory; skin; heart; eye (blink)/facial | | Psych/ physiology |

(*Continued*)

**Table 1.** (Continued)

| Publication | Description of paper | Emotional stimuli | Body location(s) measured | Self-report tool details | Subject area (psych; medic/physiol) |
|---|---|---|---|---|---|
| Lehman [87] | To assess cardiovascular functioning (BP and HR) at times of SSET and examine reports of negative emotions during these | Real life socially evaluative situations | Heart | | Psych/physiology/medic |
| Levenson [88] | Used participants' emotional facial expressions and relived emotional experiences as stimuli. Heart rate, skin conductance, finger temperature and general somatic activity were all measured as outcomes | Real life events (recalled); facial expressions (participants' own) | Skin; heart; general somatic activity | | Psych/physiology |
| Limmer [54] | Used mental arithmetic stress test as emotional stimulus, measured heart rate, pulse amplitude, skin conductance levels, electromyographic (EMG) muscle activity of the trapezius, breathing rate, and breathing amplitude, and self-reports pertaining to these locations | Stress task | Respiratory; skin; heart; eye (blink)/facial | Self-report locations with same physiological locations measured. No immediate response indicated | Psych/physiology/med |
| Lobel [89] | Investigates stress responses during computer game play, measured by heart rate | Computer games | Heart | | Psych/physiology |
| López-Benítez [90] | Emotional films were played as stimuli, with cheerfulness measured by self-report and physiological measures taken (HR and SCL); included the Discrete Emotions Scale (DES) | Films | Skin; heart; self-report (DES): gut (i.e., nausea) | DES: only composite figure reported. | Psych/physiology |
| Löw [91] | Threat and opportunity stimuli (images) presented, with physiological responses measured: heart rate, skin conductance, probe startle reflex (EMG) | Images; startle probe | Skin; heart; eye(blink)/face | | Psych/physiology |
| Madan [92] | A study of non-contact forms of heart rate measurement in emotion research | Images | Heart | | Psych/physiology |
| Mark [93] | A syringe needle was used to threaten a rubber hand, with skin conductance response measured | Rubber hand threat | Skin | | Psych/physiology |
| Marshall [94] | Using emotional faces as stimuli (NIMSTIM), along with a visual detection task, this study investigated interoceptive sensitivity, using ECG and EEG | Facial images | Heart | | Psych/physiology |
| Marshall [95] | Measured heartbeat evoked potentials and visual evoked potentials as markers of interoception in response to emotional face stimuli | Facial images | Heart | | Psych/physiology |
| Mauss [96] | Response coherence between different systems, involving emotional film stimuli and measurement of heart, skin conductance and somatic activity (i.e., body movement) | Films | Skin; heart; general somatic activity | | Psych/physiology |

(*Continued*)

**Table 1.** (Continued)

| Publication | Description of paper | Emotional stimuli | Body location(s) measured | Self-report tool details | Subject area (psych; medic/ physiol) |
|---|---|---|---|---|---|
| Meissner [97] | Emotional pictures used, measuring cardiac, EGG (electrogastrogram) and SCR | Images | Skin; heart; gut | | Psych/ physiology |
| Melzig [98] | Using interoceptive and exteroceptive threat (mild electric shock), normoventilation and hyperventilation task) in anxious participants to investigate anticipatory anxiety, with heart rate and skin conductance as physiological measures. Startle probes were used to produce defensive startle responses. Self-report on DSM IV symptoms for panic | Anticipation of shock; respiratory (hyperventilation, normoventilation), startle probe | Skin; heart; self-report (DSM symptoms): includes heart, gut, shortness of breath. Unclear if respiratory outcome can be without artifacts (i.e. how shock anticipation relates to respiratory measures). No individual symptom reporting | DSM: appears to be immediate click response after each (?) phase. Physio and self-report of heart. | Psych/ physiology/ medic |
| Mikkelsen [99] | Investigating age differences regarding links between interoceptive sensitivity (measured by Schandry) and emotional reactivity, including physiological measures (electrodermal activity and heart activity) | Images | Skin; heart | | Psych/ physiology |
| Mordkoff [100] | Emotional images, measuring HR, SCR and respiration | Images | Respiratory; skin; heart | | Psych/ physiology |
| Nair [101] | Using a stress test as an emotional stimulus to create anxiety and physiological responses (cardiovascular), with upright posture as moderator | Stress task | Heart | | Psych/ physiology |
| Noble [102] | Using EMG as a measure of stress from mental arithmetic task | Stress task | Eye(blink)/face | | Psych/ physiology/ medic |
| Notarius [103] | Stressful films, measuring heart rate, respiration rate, skin conductance, and facial expressions | Films | Respiratory; skin; heart; facial expressions | | Psych/ physiology |
| O'Brien [104] | Investigating impacts of family conflict using recordings of conflict, measuring outcomes by self-report including a tool assessing physical emotional reactions combining several responses (face felt hot or flushed; hands or body got sweaty; lump in throat and/or eyes got teary; body felt restless; heart beating faster, was pounding, or was beating louder; breathing faster; felt a rush of energy) | Audio recordings of conflict | Self-report composite: skin; heart; respiratory | Completed self-report immediately after each recording | Psych |
| Oosterwijk [105] | Using IAPS as stimuli, measured electrodermal activity and the startle response (i.e. orbicularis oculi activity) as indicators of fearful responding. | Images; startle probe | Skin; eye(blink)/face | | Psych/ physiology |
| Osborne-Crowley [106] | Investigates TBI and facial feedback for empathy, including physiological changes (skin conductance and heart rate). Self-report: HADS | Feedback from own facial expressions; feedback from own body positions | Skin; heart; self-report (HADS): includes gut ("butterflies") | HADS: used overall score only. No immediate stimulus-response | Psych/ physiology |

(*Continued*)

**Table 1.** (Continued)

| Publication | Description of paper | Emotional stimuli | Body location(s) measured | Self-report tool details | Subject area (psych; medic/ physiol) |
|---|---|---|---|---|---|
| Owens [107] | Orienting responses to IAPS stimuli, measuring cardiac activity as physiological outcome | Images | Heart | | Psych/ physiology/ medic |
| Pappens [108] | Applied resistive loads to the respiratory system for comparisons with picture stimuli, whilst acoustic startle probes (using EMG), airflow and skin conductance were used, along with subjective fear scales | Resistive respiratory load; images; startle probe | Respiratory; skin; heart; eye (blink)/facial | | Psych/ physiology |
| Park [109] | Involves sadness stimulus (a video clip) and measurement of physiological responses (heart rate, and blood volume pulse, and parasympathetic activity by assessing respiratory sinus arrhythmia), in comparisons of introversion and extraversion | Films | Heart; RSA | | Psych/ physiology |
| Pollatos [110] | Hypothesised that high interoceptive awareness will be associated with greater heart-rate reactivity and better emotional memory in a recognition task, using the Schandry task | Images | Heart | | Psych/ physiology |
| Pollatos [111] | Hypothesised that high interoceptive associated with greater heart rate deceleration, and arousal. Schandry was used to measure interoceptive sensitivity | Images | Heart | | Psych/ physiology |
| Posserud [112] | Stress tests (a Stroop and mental arithmetic test) were used to induce emotion, measured by HR, whilst distension was undertaken, in IBS patients | Stress task | Gut (rectal); heart | | Psych/ physiology/ medic |
| Price [113] | Used emotional films with respiratory sinus arrhythmia data (measured by ECG), and used MAIA and DERS (plus other tools) to collect data on interoception and emotion regulation | Films | RSA | | Psych/ physiology/ med |
| Price [114] | Used erotic images as emotional stimuli to produce startle eyeblink response, whilst in different body postures. They measured outcomes using LPPs and EMG | Images | Eye(blink)/face | | Psych/ physiology |
| Raes [115] | In a spatial cueing task, used CS (emotional faces) and US (white noise) and heart beat detection task. In one half of the sample, the latter was undertaken after the conditioning | Facial images | Heart | | Psych/ physiology |
| Richards [116] | Violent films and hyperventilation conditions were used to induce arousal, with IS measured by pulse transit time; self-report: Diagnostic Symptom Questionnaire (DSQ) | Films; hyperventilation | Heart; self-report (DSQ): DSM criteria for panic, including heart, gut, shortness of breath | DSQ: completed immediately after the task. Physio heart and self-report of heart but symptoms not reported separately | Psych/ physiology/ medic |

(*Continued*)

**Table 1.** (Continued)

| Publication | Description of paper | Emotional stimuli | Body location(s) measured | Self-report tool details | Subject area (psych; medic/ physiol) |
|---|---|---|---|---|---|
| Schäflein [117] | Comparing interoceptive accuracy (Schandry task) between people with dissociative disorder and healthy controls, and links between cardiac vagal tone (from HRV) and interoceptive accuracy. Measured sensibility using the MAIA. Facial mirror confrontation task used | Own facial reflection | Heart | | Psych/ physiology/ medic |
| Schön [118] | Used the cold pressor task, inspiratory resistive load, IAPS images, with breathing rate, heart rate and perceived dyspnoea measured | Resistive respiratory load; cold pressor task; images | (Respiratory); heart | | Psych/ physiology/ medic |
| Schweizer [119] | Using VR as emotional stimulus, measuring HR and skin conductance levels | Virtual Reality (VR) | Skin; heart | | Psych/ physiology/ medic |
| Shalom [120] | IAPS images presented to children with HFA whilst SCR was measured, along with self-reported feelings (but did not appear to include body locations) | Images | Skin | | Psych/ physiology/ medic |
| Stemmler [121] | Tasks and imagery used, measuring self-report (limited number, generic), facial movements, somatic movements, heart rate, respiration rate, SCR and body temperature | Stress task; images | Respiratory; skin; heart; eye (blink)/face; general somatic movements | | Psych/ physiology |
| Stephens [122] | Music and film clips used, measuring cardiac, respiratory systems and SCR | Films; music | Respiratory; skin; heart | | Psych/ physiology |
| Sternbach [123] | The film "Bambi" used, measuring skin resistance; gastric motility; respiration rate; heart rate; eyeblink rate and finger pulse volume | Films | Respiratory; skin; heart; gut; eye (blink)/face | | Psych/ physiology |
| Sze [124] | Whether people with higher IA have greater emotional coherence between subjective emotional experience and physiological responding. Meditators compared with non-meditators and dancers. Heart period measured. Self-reported visceral awareness measured using several scales (Autonomic Perception Questionnaire, APQ, Body Consciousness Questionnaire, BCQ) | Films | Heart; self-report (APQ and BCQ): includes heart, gut, face, respiratory | APQ and BCQ: completed before the study | Psych/ physiology |
| Tajadura-Jiménez [125] | The impact of sound (heart beat sounds) and IAPS on physiology (heart beat) and emotional experience | Heart beat sounds; images | Heart | | Psych/ physiology |
| Tsai [126] | Asked participants to recall emotional events in their lives, as emotional stimuli, measuring facial expressions as responses and physiological responses: cardiovascular, electrodermal, and respiratory systems | Real life events (recalled) | Respiratory; skin; heart; facial expressions | | Psych/ physiology |

(*Continued*)

**Table 1.** (Continued)

| Publication | Description of paper | Emotional stimuli | Body location(s) measured | Self-report tool details | Subject area (psych; medic/ physiol) |
|---|---|---|---|---|---|
| Tsai [127] | Emotional films as stimuli, measured skin conductance level (SCL), heart activity, finger temperature, and respiratory activity (time between inspirations) | Films | Respiratory; skin; heart | | Psych/ physiology |
| Uchiyama [128] | Real life emotional stimulus in lab, measuring heart activity, GSR and respiration rate | Real life events (lab-based situations) | Respiratory; skin; heart | | Psych/ physiology |
| Van Den Houte [129] | Using negative images to induce emotions in FS patients, measuring by self-report: tight feeling in the chest, heart pounding, stomach ache, headache, fatigue, difficulty breathing, faster heart rate, nausea, dizziness, and muscle ache. Heart rate, skin conductance levels, and fractional end-tidal $CO_2$ were all measured throughout | Images | Skin; heart; respiratory; self-report: includes stomach ache and difficulty breathing, but not reported separately | Heart and respiratory measures and self-report of same locations. No immediate response apparent | Psych/ physiology/ medic |
| Van Oyen-Witvliet [130] | Used emotional sentences as stimuli, measured SCR, eyeblink (EMG) with startle probe, and HR | Written sentences | Skin; heart; eye(blink)/face | | Psych/ physiology |
| Vianna [131] | Used film clips, measuring HR, SCR and EGG (electrogastrogram) | Films | Skin; heart; gut | | Psych/ physiology |
| Vrana [132] | Imagined scenarios, measurements of SCL, heart activity, EMG | Imagined imagery | Skin; heart; eye(blink)/face | | Psych/ physiology |
| Weinreich [133] | Investigating the effect of an emotional picture task (rating album covers) on EMG | Images | Eye(blink)/face | | Psych/ physiology |
| Werner [134] | Uses public speaking as emotion stimulus in a comparison of people with high IS and low IS, including physiological measures (HR, skin conductance), and impacts on anxiety. Used the Questionnaire for Speaking Anxiety (QSA) to measure physical symptoms (including heart and gastric sensations) | Public speaking | Skin; heart; self-report (QSA): includes heart and gut, but individual symptoms not reported | QSA: completed at the time of imagining public speaking | Psych/ physiology |
| Werner [135] | Comparing those with high or low interoceptive awareness (assessed by the Schandry) on affective responses to provoked social exclusion. ECG and skin conductance measures taken | Social exclusion situation | Skin; heart | | Psych/ physiology |
| Winton [136] | Emotional images, measuring HR, SCR and externally rated facial expressiveness | Images | Skin; heart; facial expressions | | Psych/ physiology |
| Yao [137] | Using oxytocin along with emotional face stimuli, participants had to indicate when they detected their heart beat | Oxytocin (emotion hormone); facial images | Heart | | Psych/ physiology |
| Zuckerman [138] | Used emotional films, measuring heart activity, SCR, and rating facial expressions | Films | Skin; heart; facial expressions | | Psych/ physiology |

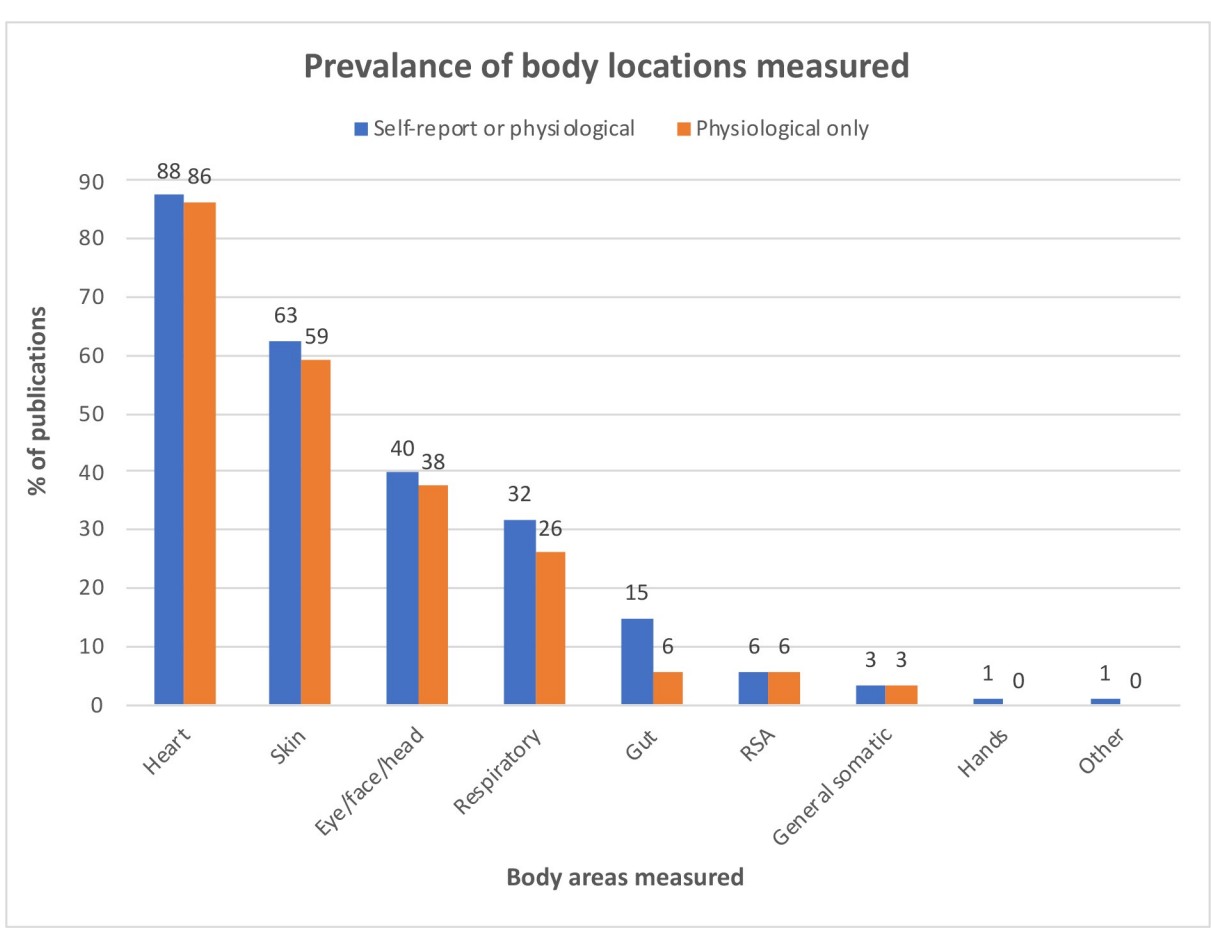

Base: 88 publications.

**Fig 3. Summary of locations measured in general texts.**

Jung, Ryu, Lee, Wallraven & Chae [139] used "a bodily sensation map-emotion (BSM-E) application" (p. 5). Participants could similarly report where sensations were felt in their bodies and could leave areas blank where nothing was felt, but this tool did not appear to record levels of deactivation, only varying levels of activation. These authors also created "Regions of Interest" (ROI) formed from clusters of clicks, which formed part of their analysis.

**Overview of body mapping studies.** Overall, the core texts utilizing body mapping technology used a range of emotional stimuli: images, GIFs, music, films, words, and sentences. Two report having undertaken studies in English [141,146], including a cross-cultural study [147], one was carried out in an English-speaking country but without an explicit statement on the language used, two reported their procedures in English, including quotations of experimental materials delivered to participants, but in nations where English is not the native language [139,145], with the remainder having undertaken studies in Finnish [140], or Finnish, Swedish and Taiwanese across a sample [50]. All studies used emotional stimuli consistent with basic emotion categorisations (i.e., anger, fear, surprise, happiness, disgust and sadness), with some also including non-basic/complex emotions (i.e., anxiety, depression, pride etc.), and one using stimuli pertaining to social scenarios [145]. Two publications [141,145] also included distinctions between self and other, i.e., separate tasks focusing on first-person responses to emotional stimuli and another involving judgements of the mapped responses of

**Table 2. Core texts.**

| Publication | Description of paper | Sample | Methods | Main findings | Emotional stimuli | Body location(s) measured & reported | Time limit |
|---|---|---|---|---|---|---|---|
| Bergquist [142] | Investigated the bodily symptoms of stress in people with drug and alcohol dependency. Sought to identify connections between stress and drug/alcohol related responses, and whether self-report is consistent with previous physiological studies | n = 56 cocaine/alcohol dependent participants | Used a scene construction questionnaire (SCQ) that included an emotional and physiological response checklist with people with drug and alcohol dependence. Participants recalled stressful and drug-related events then completed a checklist of 67 bodily sensations they recalled having experienced during the situation | Most likely body locations to be reported by participants as active in the stress condition: heart = 91.1%; respiratory (breathing) = 78.6% and gut (stomach) = 71.4%, followed by skin (sweating) = 67.9%, consistent with general arousal from autonomic activation, consistent with previously reported laboratory autonomic, neuroendocrine, and subjective responses | Scene Development Questionnaire (SCQ) involving recalled stressful scenarios | Self-report (chosen from list of 67 recalled sensations): including: respiratory, heart, gut, skin, face, eyes, whole body/general somatic | No limit |
| Hietanen [140] | A study of body maps in response to emotion words in children of various ages (i.e., memories of feeling each emotion type) to see how these differ over time | n = 48 preschool; n = 68 2nd grade; n = 91 4th grade; n = 42 8th grade, and n = 46 high-school children/adolescents. Comparison data also taken from adults (n = 36) | As in Nummenmaa [50], this study used written emotion words as stimuli (to facilitate recall) for emotion types (6 basic emotions in this case), involving children and adolescents. This was a pen and paper version of the original. Overall, the method was the same as previously, with pen and paper responses transferred to the emBODY tool to create the heat maps | Despite finding somewhat discrete body maps, these were more similar to each other in the case of children compared with adults. The set of child body maps begin to look more like the adult set the older the child participants. Whilst happiness and surprise were found to have emerged in the 6 year old group, disgust was the last to appear similar to that of the adult group | Written emotion words (in Finnish) to promote recall (also read out loud to 6 year old participants) | Self-report: open ended, on body map | No limit |
| Hubert [143] | The effects of positive and negative emotional films regarding correspondence between subjective reports and physiological response, and anxiety disorder. Sample of anxious vs. low anxious participants. Authors predicted activation in all measured body parts; greater correspondence between subjectivity and objectivity for anxious versus controls | n = 24 (12 with generalised anxiety disorder; 12 matched controls, matched by age, sex and educational level) | Following positive and negative film stimuli (two films lasting 9 mins each: "Raiders of the Lost Arc" and "Peanuts", which were expected to produce several affective states), participants self-rated on a Visual Analogue Scale (VAS) for 14 items, immediately before and after the end of each film. The body location items pertained to: the heart, respiratory system, the gut (i.e. tense stomach), skin, eyes (i.e., tears), face (i.e., hot face), head (i.e., rush of blood). Concurrent EMG, skin conductance level (SCL), respiratory activity and ECG measures. Self-ratings of mood (e.g., anxiety). | Activity in heart, respiratory system & gut were prominent self-reports. Negative films: "heart rate increases", "faster breathing", "tense stomach"; positive films: decreased "tense stomach". EMG (corrugator): increased activity for negative, decreases for positive. EMG (zygomaticus): increases for positive. Respiration: increases from both stimuli. HR decreased during negative, SCL increased during both stimuli. Anxious & low anxious group correlations: anxiety & "heart rate increases", "tense stomach", "difficulties breathing" ("shallow breathing" for low anxious group). Correlations between subjective & objective measures in the anxious group only | Films | Physio: respiratory, heart, skin, eye (blink)/face. Self-report: respiratory, heart, skin, eye/face, gut, head | No clear limit (responses immediately after a 9-minute film) |
| Hubert [144] | The aims of the paper are similar to the authors' earlier paper [143], but focussing on a healthy sample, and without measurement of respiration or EMG. The authors investigated whether subjective and objective responses would be enhanced after suspenseful films, as opposed to showing reduced responding during pleasant films | n = 20 (male, non-psychology students) | Seemingly, the same film stimuli were used as in the previous study, but lasting for 10 mins each. The same VAS items were used as previously. Heart rate, skin conductance level, spontaneous fluctuations were measured. Heart rate and skin recordings were taken in 60 second periods. All other procedures were fundamentally the same as in the 1990 paper [143] | In a low-anxious only sample, this study showed overall changes in 'hot face', 'sweating', 'heart rate increasing', 'sweaty palms', 'restlessness' and 'difficulties in breathing'. For 'suspense' films, increases in all of these, but for pleasant films: increases in 'restlessness'. Physiological changes (film x time interactions): skin conductance and SF. For suspense and pleasant: heart rate decreases. For skin conductance: increases in suspense, decreases during pleasant films | Films | Physiological measures: heart, skin. Self-report: respiratory, heart, skin, eye/face, gut, head | No clear limit (responses immediately after a 10-minute film) |

*(Continued)*

**Table 2.** (Continued)

| Publication | Description of paper | Sample | Methods | Main findings | Emotional stimuli | Body location(s) measured & reported | Time limit |
|---|---|---|---|---|---|---|---|
| Jung [139] | A study of whether interoceptive accuracy influences topographical body maps, differing by emotion type. Hypothesized that higher interoceptive accuracy would be associated with stronger sensations in emotion-specific bodily locations | n = 31 (15 female; mean age: 24.1 ± 4.5 years) | Provoked anger, fear, disgust, happiness, sadness, and neutral. The emotional nature of the task was disguised using a visual accuracy task, a high difficulty "masking task". For map drawing, the authors did not use the emBODY tool, but "a bodily sensation map-emotion (BSM-E) application". Participants reported where sensations were felt leaving areas blank where nothing was felt. "Regions of Interest" (ROI) were identified as emotionally relevant clusters: "circular area with a radius of 10 pixels from the peak points of significant bodily sensation clusters" | All emotions were represented in the chest, which is consistent with the other findings, and again anger was linked to hands (as was fear, which was also active in lower legs and feet). Disgust was active along the gastrointestinal tract. The average magnitude of sensation in the regions of interest (where responses clustered) was positively, moderately, correlated with performance on the Schandry task (r = .367, p = .042). The authors conclude "These findings suggest that individuals with more accurate interoception had stronger sensations in emotion-specific bodily locations" | GIF images, facial images | Physiological measures: heart rate detection (Schandry task). Self-report: open ended, on body map | Each trial = 20 seconds long, followed by 30 seconds observing bodily changes, and 150 seconds responding on body maps |
| Novembre [145] | Hypothesised dissociation between negative and positive social scenarios. Self vs other comparisons, i.e., overlap between conditions. Higher arousal for the self condition; same valence similarity between social scenarios and basic emotion trials | n = 91 (65 females; mean age = 30.1; SD = 9.1) | Online study, using eight matched negative & positive social scenarios (bereavement/birth, romantic rejection/acceptance, exclusion/inclusion, negative/positive evaluation). Participants indicated sensations on body maps. Stimuli: written statements for each category; e.g. "the person you love leaves you". Repeated Nummenmaa et al [50], with each emotion at the centre of the screen. Authors divided body maps into head, chest, abdomen, upper limbs (arms and hands), & lower limbs (legs and feet), averaging intensity values in each area | After imagining social scenarios, the face, head & chest were active in response to positive and negative stimuli; with deactivation in arms and legs following negative. For Nummenmaa replication: sadness presented significant deactivations, in arms and legs. Other emotions were active in the head (not fear) and chest (not disgust). Anger showed activation in the arms and hands; happiness active in the chest, arms, upper abdomen. . .". . .activation of the chest and face areas and deactivation of the limbs represented the most consistent pattern. . .". Activation in upper limbs for positive scenarios, some localised responses (e.g., sadness around the eyes) | Written statements | Self-report: open ended, on body map | No limit |
| Nummenmaa [50] | Images, films, stories used to induce or recall emotions with participants indicating on body maps (using the "emBODY" tool) where they felt these, against a backdrop of basic emotion theory | n = 773 (experiment 1a (Finnish): n = 302; experiment 1b (Swedish): n = 52; experiment 1c (Taiwanese Hokkien): n = 36; experiment 2: n = 108; experiment 3: n = 94; experiment 4: n = 109; and experiment 5: n = 72) | Five experiments (n = 36–302), including a cross-cultural sample (Western European & East Asian), seemingly using only stimuli in the first language of participants. Expt 1: emotion words onscreen for 6 basic & 7 non-basic emotion types, neutral trial. Expt 2: vignettes with "online" body mapping. Expt 3: films with "online" body mapping for fear, disgust, happiness, sadness, neutral. Expt 4: facial images to test for emotion recognition in others, with body maps (anger, fear, disgust, happiness, sadness, surprise). Expt 5: categorised the body maps of others | Distinct body maps for each emotion, similar maps in each category regardless of emotional stimulus (Expts 1–3). Authors conclude that emotion words function similarly to non-verbal stimuli regarding their ability to elicit body maps of subjective sensations. Authors proposed that consistency of body maps indicated universality for emotion types. Expt 4: concordance between body maps and those in the other expts. Expt 5: 46% mean overall accuracy (vs. 14% chance) when categorising maps produced by others. Throughout expts, there were lower but significant correlations between mismatching emotion types (e.g. between happiness and anger). High concordance across experiments | Written emotion words, stories, films, facial expressions | Self-report: open ended, on body map | For films and stories, participants reported "online" (i.e. during the stimuli). No time limits imposed. Film stimuli lasted for 10s each, but participants were encouraged to view as many times as required |

(*Continued*)

**Table 2.** (Continued)

| Publication | Description of paper | Sample | Methods | Main findings | Emotional stimuli | Body location(s) measured & reported | Time limit |
|---|---|---|---|---|---|---|---|
| Sachs [141] | Measuring self-reported sensations for self and other in adults and children on body maps. The authors investigated whether these were related to trait empathy and age | First study: n = 82 (59 females, mean age = 22.20, age range: 18–27, SD = 1.62); second study: n = 60 (children aged 8–11; 26 females, mean age = 9.93, SD = 0.57) | Film clips (90 secs each) and music intended to provoke one of 4 emotional states; empathy-appropriate (for film: happiness, sadness, fear, mixture of emotion; for music: happiness, sadness, anger, calmness), participants indicating sensation on body outline, on another where the film character/musical performer felt sensation. Cognitive & affective empathy was measured. Participants selected which emotion they/other felt most strongly, and how intensely. 2nd study (children): similar procedure using paper and pencils | For the first study: "Visual inspection of the averaged body maps appear to match colloquial and cross-cultural understanding of emotions and where they are felt on the body, such as in the head and on the chest (Nummenmaa et al., [50]). Participants also reported more activity on the body during higher arousal emotions, i.e. anger and happy, and less activity on the body in response to low-arousal emotions, i.e. sadness and calm". In the second study: children showed less ability to match their own personal maps with that of the 'other', compared with adult participants, suggesting that their ability to apply their own case to others is under development | Films, music | Self-report: open ended, on body map | No clear limit (appeared to produce body maps during stimuli presentation— either music or film clip lasting, on average, 90 seconds) |
| Torregrossa [146] | Tested the hypothesis that embodied emotion would be disrupted in people with schizophrenia, regarding the interoceptive and somatosensory contribution to self | n = 26 in each of two matched (by age and sex) groups: schizophrenia & control | Emotion word stimuli: "neutral", "fear", "anger", "disgust", "sadness", "happiness", "surprise", "anxiety", "love", "depression", "contempt", "pride", "shame", and "jealousy". Definitions provided for each word. Participants coloured in two silhouettes, one to indicate where activity was increasing, and another where decreasing, then combined to create body maps | Independent body maps found for each emotion type in the control condition. Indistinguishable body maps for schizophrenia group, particularly for low arousal emotions (sadness, depression), with the highest correlations between groups being for high arousal emotions (love, pride, fear). Overall, controls showed greater activation compared with schizophrenia, particularly for high arousal emotions; schizophrenia group showed more deactivation for high arousal emotions. | Written emotion words | Self-report: open ended, on body map | No limit |
| Volynets [147] | A cross-cultural study involving participants from across 101 countries measuring self-reported sensations using body maps | n = 3,954 (3,260 females; 101 countries, aged 18–90; mean = 34.9 years) | Grouping countries by major civilization (and at times by Western vs. non-Western), this study followed previous studies (including Nummenmaa et al [50]) in presenting an emotion word (6 basic, 7 complex and one neutral) between silhouettes, indicating felt sensations (one for activation, one for deactivation) in an online study | Authors state that the maps have revealed universal sensation patterns for each emotion type. Age: absolute intensity scores negatively correlated with age (rs = 0.11, p < .001). Western vs. non-Western: Western participants reported more activation for fear, anxiety, disgust, happiness, love, pride, contempt, jealousy, anger, and while being neutral, with non-Western participants reporting less deactivation for depression, sadness, and shame, (t > 2.65, p < .05, FDR corrected). Females reported more activation in their "guts" during anger, jealousy, anxiety and shame, and throat during anxiety, shame, fear, contempt, and sadness (plus more activation in the heart area) | Written emotion words | Self-report: open ended, on body map | No limit |

a second-person. None of these mapping studies required speeded responses within restricted timescales, which is perhaps a reflection of the focus on basic emotion categories, which may take time to unfold as multi-componential syndromes. Only one mapping study [139] undertook bodily/physiological measurement (the Schandry task), which was also the only core text to assess interoceptive sensitivity. Further, whilst the body mapping tool could be viewed as an attempt to measure this construct, no study assessed interoceptive sensibility using an established tool, such as the MAIA [6]. Hence, in general, measurement of interoception was overlooked.

**Overview of non-mapping studies.** Of the remaining texts, two reported studies by the same authors [143,144] using the same self-report tool: a visual analogue scale (VAS) in combination with a 14-item list of bodily symptoms, including bodily locations. These symptoms had nuanced qualities; for instance, the list contains specific references to "shallow breathing", "tense stomach", "heart rate increasing" and "sweaty palms", as opposed to more generic indications of bodily changes with wider applicability. In turn, participants may have been less likely to report these more specific outcomes (e.g., noticing their heart rate, but not necessarily as "heart rate increasing"). Overall, the list of self-report items included the heart, respiratory system, the gut (stomach), the head/face (including the eyes/crying), the skin (e.g., sweating) and general somatic activity (e.g., trembling). Both studies recorded physiological changes after emotional stimuli were applied: one, a study of anxiety [143], recorded facial/eye changes via EMG, skin conductance, heart activity by ECG, and changes in respiration by a thermistor. The other study recorded only skin conductance and heart activity [144]. These were the only core texts to measure multiple physiological systems in combination with self-report. The emotional stimuli used were films (seemingly the same films across the two publications), which were intended to be either generically pleasant or unpleasant [143], or positive or suspenseful [144]. Again, no speeded responses appeared to have been required of participants within restricted time scales, with the film stimuli lasting for up to 10 minutes in each instance followed by self-reports within an undisclosed period. As was mostly the case in the wider context of the general texts, besides self-report options there was no reporting of gastrointestinal activity at the bodily/physiological level in either publication alongside the cardiac, skin, EMG and respiratory measurements. Finally, there is some crossover with the body mapping studies in terms of basic emotion categories, with some reporting on "mood", e.g., disgust and sadness, although these were not indexed to the stimuli themselves.

The one remaining study [142] focussed on stress, drug and alcohol dependency using a Scene Construction Questionnaire to develop stimuli, with participants recalling stressful and drug/alcohol scenarios (we focus on only the former here). Participants reported any bodily changes from a list of 67 sensations, involving multiple body areas relating to physical symptoms at general levels (e.g., "Heart") and broken down into more specific symptoms (e.g., "Heart pounds", "Heart beats faster", "Heart skips a beat" etc.). The majority of body locations are covered in the list of responses: head/face (including eyes/crying), throat, chest (heart and respiratory), gut (stomach), arms, hands, skin and overall somatic activity. No physiological recordings were taken, with all body-related data consisting of reported symptoms, again without a time restriction. Finally, unlike the mapping studies, there was no framing of the stimuli in line with basic emotion categories. Whilst outcomes from the list of sensations included some basic emotions, these were not systematically combined with body sensations to indicate multi-componential syndromes pertaining to those categories.

**Body mapping findings.** The body mapping studies tended to identify distinct maps for each emotion type (and for social scenarios [145]. These are suggested to be patterns of "net sensations" for each type [50]; i.e., a subjective summation of bodily changes in multiple areas during an emotional response. There were exceptions to this distinctiveness, with sub-groups of participants displaying reduced differentiation between emotion types, i.e., those with

schizophrenia, especially regarding low arousal emotions [146], child participants [140,141], older participants [147] and, in the case of schizophrenia and older age, reduced levels of activation, even for high arousal emotions. For the single mapping study that included a physiological component [139] the resulting Schandry task scores were, overall, positively correlated with the magnitude of activation indicated in the ROIs, which may suggest that greater sensitivity results in stronger bodily sensations pertaining to emotion. The ability to correctly categorise the body maps of others was also found [50], and consistency between self/other maps [141,145]. In addition to the distinctiveness of maps for each emotion type–both within and between publications–these also displayed a level of consistency of activation/deactivation *across* types (e.g., [147]), as well as some inter-study inconsistencies regarding single emotion types across publications (see assessment below, Table 3).

*Visual inspection of body maps.* Since each map is intended to display a representation of a "net sensation", i.e., a summative subjective report, it should be possible to visually inspect each of these to arrive at conclusions on which body areas were deemed to be reliably activated or deactivated (or neutral) during an emotion. That is, the 'heat' (activation), 'coolness' (deactivation) and unchanged areas of heat maps putatively correspond to the 'heat' or 'coolness' of an inter-subjective embodied experience. Indeed, this is in large part the value of such a visualization, in addition to the statistical inferences that can be undertaken using pixel selection data. For this reason, we have summarised an attempt at visual inspection (see S2 Table, and Table 3) as activation (A), deactivation (D) and neutrality (N) in areas for each emotion type, for each publication. This is intended to allow comparisons and contrasts between similar mapping studies regarding which body areas change in emotional responses. From there, the degree of consistent identification is indicated per body part (with A or D indicating a reported bodily change, and N no change), across body mapping texts, drawing on only the investigation of basic emotions (i.e., not complex emotions or social scenarios) from healthy adult samples, to allow comparability. Although basic emotion categories themselves are not the intended focus of this review, we present the results of these papers in terms of the authors' own categorisation. In any case, this may indicate common changes across emotion types, suggesting perhaps more fundamental body locations, which may also be foundational; i.e., always occurring during a response, regardless of duration or complexity.

To undertake this visual assessment, we have made bodily distinctions based, in part, on those made within the texts themselves: (i) head/face (changes anywhere within the boundary of the head outline, including eyes, cheeks etc.); (ii) chest (encompassing the area from the termination of the throat down to the horizontal line that cuts through the elbows on the body silhouettes); (iii) the throat (the area from the termination of the upper chest to immediately below the jawline); (iv) the upper arms (including the shoulders, down to the inward curve of

**Table 3. Summary of body mapping consistency of location.**

| Proportion of basic emotions with consistency | Body location assuming (T) = genuine trace of change | Body location assuming (T) = subjective 'noise' |
|---|---|---|
| 6/6 | Head/face | N/A |
| 5/6 | Throat; chest; lower abdomen | Head/face; chest |
| 4/6 | N/A | Throat; groin (consistently neutral); upper arms; upper legs |
| 3/6 | Upper arms; hands; groin (consistently neutral) | Lower legs; lower arms; hands (2/6 = consistently neutral) |
| 2/6 | Lower arms | Feet; lower abdomen |
| 1/6 | Feet; upper legs; lower legs | N/A |

the elbows); (v) the lower arms (from the inward curve of the elbows to the base of the hands); (vi) hands; (vii) lower abdomen (from the horizontal line cutting through the elbows to the horizontal line cutting through the base of the hands); (viii) the groin (from the base of the lower abdomen to the horizontal line cutting through the top of the legs); (ix) upper legs (from the base of the groin to the inward curve at the top of the knee); (x) lower legs (from the top of the knee to the termination of the inward curve of the ankle); (xi) the feet.

In some cases, the maps displayed only a trace change tending towards neutrality as opposed to clear shifts in subjective experience. Relevantly to this, Volynets, Glerean, Hietanen, Hari & Nummenmaa [147] state the need to consider the vast array of possible influences (including cultural) on emotional responding that inevitably leads to a degree of mapping variability (which they otherwise consider to be genuinely universal). Readers are, thereby, encouraged to focus more on the degree of commonality across contexts as opposed to relatively minor differences: "...our view of psychological universality refers to the degree of consistency of psychological phenomena across cultures, rather than naïve all-or-nothing conceptualization" ([147], p. 8).

Therefore, in cases of trace change, the "T" is used to indicate a level of ambiguity between a trace of subjective experience and purely uninformative subjective noise. This then retains an observable (but minor) distinction between maps without forcing a conclusion on consistency due to a potentially arbitrary judgement. Further, we make no other distinctions regarding differences in the level of clear activation or deactivation (i.e., that do not approach a value of N). Finally, we have allowed for one discrepancy from the majority of texts per body part (i.e., one of the seven texts in each case). For example, if one body map for an emotion type displays neutrality with the remaining publications showing agreed activation in a body part, this remains classified as consistency. In this way, having taken ambiguity into account and allowing for a minor degree of error, any remaining discrepancies after implementing a relaxed standard may suggest a more significant departure from consistency.

Across the options, the head/face, chest and throat show the greatest degree of agreement throughout emotion types regarding changed activity following an emotional stimulus. The peripheral body parts show the most 'traces' of sensation. With the exception of the lower abdomen (where displayed changes were for the most part either a potential spill over from reported changes in the chest or were otherwise barely observable on the maps), it is the peripheral body parts that are less likely to show agreement across publications. (The groin was only consistently neutral, hence less relevant to this categorization.)

If taking a simple unweighted average of the proportion of consistency across the two interpretative options, the results indicate an order of consistency that broadly reflects the distance of each body part from the head/face, a fact reflected under each option:

1. Head/face (5.5).

2. Chest (5).

3. Throat (4.5).

4. Upper arms; lower abdomen (3.5).

5. Lower arms; upper legs (2.5).

6. Hands (subtracting the two neutral cases); lower legs (2).

7. Feet (1.5).

Regarding each emotion type, the greatest degree of agreement regarding A, D or N, and where "T" is considered to be a genuine trace of activity, was found in anger and sadness (7/11

body areas), followed by happiness and surprise (6/11), fear (5/11) and disgust (3/11). Alternatively, where "T" is interpreted as 'noise', surprise alone showed the greatest degree of agreement (10/11 areas), followed by disgust (9/11), anger (8/11), fear (6/11), happiness (4/11) and sadness (1/11).

**Non-body mapping findings.** The full results for active body locations reported by the three more traditional self-report studies are provided in S3 Table. Comparisons between these and the body mapping studies are limited by several factors, not least of which is the limited numbers of body areas reported on (i.e., 6 of the original 11), the use of more general emotional stimuli (i.e., "stress", "pleasant", "suspenseful etc.) in addition to contrasts between the self-report tools used.

*Visual analogue scale studies.* In the study of anxiety [143], self-reports indicated the head/face ("blood rushing to the head"), heart, respiratory system and gut to be active in response to stimuli (with the gut self-reports indicating increasing or decreasing tension depending on whether stimuli were negative or positive, respectively), along with changes in skin conductance, changes in EMG and heart rate. Self-reported disgust was found to correlate with a "tense stomach", "shallow breathing" and a negative correlation of disgust with physiological measures of respiratory activity (in the high anxiety group). For self-reported anxiety in both the highly anxious and low anxious groups, there were accompanying self-reports of "heart rate increasing", changes in breathing and "tense stomach"; for the highly anxious group only, there were corresponding changes in actual respiratory activity. For "joy", the highly anxious group self-reported respiratory changes. There were also subjective-objective correlations between skin conductance measures and self-reports of "blood rushing to the head", "heart rate increasing", "breathing faster" and "tense stomach" (the high anxiety group only). The other study by the same authors [144] has similar self-reports of heart, respiratory, and facial changes but no gut changes, along with actual heart changes (ECG) and sweating palms, in response to "suspenseful" film stimuli.

*Scene construction questionnaire study.* Finally, the SCQ study [142] reported a long list of sample proportions for the symptoms of "stress". These symptoms related to the head/face, chest, throat, arm, hands, and lower abdomen, with a predominance of heart (91.1%); respiratory (breathing; 78.6%); gut (stomach; 71.4%); a cluster of head/face reports, including tightness in the face (55.4%); gritted teeth (55.4%); forehead tension (53.6%); and tears (50.0%).

**Overview of core text findings.** The connections between the 10 core publications consist, then, in the widespread findings on: (i) the head/face (by self-report supported by findings from EMG, in particular); (ii) the chest in general, with specific findings on the heart (by self-report and ECG); (iii) specific findings on the respiratory system (self-report and thermistor recordings); and (iv) support for activation of the gut (but by self-report only, and which shows limited consistency across the body mapping studies). Self-reports on the throat, too, may indicate gut changes, but which may also be interpretable as respiratory changes, or both. Overall, some of the more pervasive changes displayed in the maps, such as in peripheral body parts, could relate to the correspondence found in the two "VAS" studies between skin conductance changes and self-reported activation in the head/face, heart, respiratory system, and to a lesser degree, the gut/stomach. Further, in the case of basic emotion categories, where these were reported in relation to body locations in the traditional self-report studies, self-reported disgust indicates some consistency regarding the gut (stomach) between the mapping studies and the study of anxiety (but less obviously so for the chest/respiratory system), for anxiety/fear regarding the heart (self-report) and the respiratory system (self-report and physiological), and for joy/happiness regarding self-reports of the chest/respiratory system. Finally, there was also the limited support from the few general texts cited (i.e., for subjective-objective concordance for the heart and skin).

## Discussion

The findings from the wider collection of publications confirms the predominance of the heart in research involving measures of bodily changes in the investigation of human emotion. The respiratory system was perhaps not as widely investigated as anticipated, although many studies were excluded due to artefacts (e.g., an inspiratory load was the emotional stimulus, activity in the respiratory system was an outcome measure). Regarding the one remaining member of the visceroceptive trio–the gastrointestinal tract–in only a minority of cases was there measurement by self-report or by a physiological tool such as the electrogastrogram (EGG).

### Lacking measurement of the gastrointestinal system

As noted by the authors of one of those cases [131] the gut has been relatively under-researched in emotion studies ("In psychophysiology, measures of electrodermal activity, cardiac function, facial EMG, and respiration have been used frequently to assess the emotional states of experimental participants. . .Remarkably, the gastrointestinal system has been almost completely neglected. . .", p. 70), which is despite the non-invasiveness and reliability of EGG (p. 71). Since publication of that paper, the situation has not vastly improved. This general lack of emphasis on measuring all three visceroceptive domains, which appear to be of particular relevance to embodied emotion, means that potentially useful findings on emotion regulation are being missed. With an accompanying scarcity of studies using *both* self-report and physiological responses, we found limited consideration of the concurrence of subjectivity and objectivity in emotion research. This was also true of the core texts, with only three instances of concurrence, again without objective measurement of the gut despite all 10 texts presenting the opportunity for gut-related self-reports. If there is an aim to investigate the bodily contributions to emotional experience, it is necessary to rely on some sort of self-report or subjective display, and to have this grounded somehow in factual bodily changes. This aim is consistent with the tripartite model of interoception [5] and embodied emotion research [1,2] more generally.

### Lacking measurement of interoception

Relatedly, without physiological measurement there can be no measurement of interoceptive sensitivity, part of the tripartite model. The one exception among the core texts [139] used the Schandry task [11], which has been the target of recent criticism, with charges of pervasively undercounted heartbeats [148], and that the task may be influenced by pre-existing knowledge of heartbeat [149–151]. Indeed, in the case of Jung, Ryu, Lee, Wallraven & Chae [139], the reader is provided with only a scatterplot, without raw Schandry scores. If scores are substantially higher than the mean usually found in interoception studies that categorise participants as highly interoceptive (e.g., [110]), this could indicate improvements in performance via non-interoceptive influences. The remaining core texts measuring physiology were not overtly concerned with interoception (and, indeed, were published some time before the more recent explosion of interest in the construct).

A key finding of the review, therefore, is that–for the most part–studies of emotional feeling and body location were not grounded in interoceptive research, under any articulation. The body mapping authors variously refer to interoception (e.g., [139,142,144,146]) but, even where it was possible to measure the construct as articulated in the literature [5,6] there was little attempt to do so. Also, given that the body mapping tools have not been reported as validated using established interoceptive measurements (although one study [139] may be seen as an attempt to do so), there is the potential to undertake validation not only in relation to sensitivity measures, but also with established sensibility measures such as the Body Perception

Questionnaire [152] and other articulations of interoception, such as the MAIA [6]. Another consequence of having no measures of sensitivity and validated sensibility is that a calculation of interoceptive awareness (i.e., a metacognitive construct [5]) is not possible. Whilst it cannot be assumed that sensitivity measurements will map onto sensibility measurements (e.g., [5]), any possibility of detecting correspondence evaporates if studies do not attempt to collect such data. It may be particularly important to gather data across different interoceptive channels, which may be a decisive factor if interoceptive sensitivity is indeed indexed to specific channels.

## Using independently validated stimuli

Further, the wide variety of stimuli used across the core studies were for the most part adopted or developed by the authors themselves, thereby ensuring they reflected the basic emotion category each stimulus was intended to elicit. As indicated previously, there are established stimulus sets (e.g., the IAPS), which have previously undergone assessment of their ability to produce detectable physiological responses. Hence, whilst the mapping in particular is an innovative approach to self-report, it may be worthwhile using it with stimuli that have an *a priori* likelihood of producing physiological changes to thereby increase the likelihood that the body maps genuinely represent underlying physical changes. If there can be no accompanying physiological measurement, there would then at least be a stronger implication of objectivity. In addition, without independent validation, there is the risk that stimuli could inadvertently help to articulate a desired outcome. Indeed, this may relate to the measurement of basic emotions themselves.

## The emphasis on basic emotions

With the stimuli being categorised as they are in many of the core texts, these may have the potential to act as primers for basic emotion outcomes. The majority of core papers were concerned with basic emotions, again with the body mapping papers in particular tending to categorize stimuli accordingly. In some cases [50,140]), there was yet more explicit grounding in basic emotion theory, with assertions that basic emotions have been established neurophysiologically, with the role of the body maps being to confirm this reality at a subjective level. However, the basic emotion view is controversial, with much in the way of counter-evidence.

There is, for instance, a research cluster involving the work of Lisa Feldman Barrett where, rather than conclusions on discrete emotion types, the evidence points to more fundamental biological features that interact with constructive processes to create emotional instances [153,154]. If the latter model is true, there would be no requirement for discrete bodily activation to match discrete subjective body maps.

## Concept-based vs sensation-based interpretations of the body mapping texts

This latter suggestion pertains in particular to the verbal/linguistic stimuli used in the body mapping studies, including emotion words and statements, where in one case participants were provided with definitions of the words used [146]. Three of those seven studies were likely undertaken in English, including the cross-cultural study [147]. Whilst it was recognised that the body maps may be concept-based, culture-independent readouts [50], authors indicate that their use of multiple stimulus types, principally "non-verbal" stimuli (e.g., films without words), resulted in comparable body maps to those elicited using words. The argument here appears to be, then, that since there is parity between non-verbal and verbal stimuli this indicates that the sensation-based reports are not limited to concepts encapsulated in language.

However, there is also the possibility that this demonstration of parity between stimulus types illustrates that "non-verbal" stimuli draw on the same concept-based, culture-independent readouts as the explicitly linguistic stimuli. Hence, this may be why it was possible to provide a participant with an emotion word (e.g., "disgust"), which did not involve inducing emotion [50], and thereby find a similar map to the one produced when showing a disgusting film. To some extent, then, the stimulus types may articulate responses on the maps due to a shared "readout" or script. The sense of this suggestion is seen, for example, in maps that highlight the eyes in "sadness", the hands in "anger", and the whole body in "happiness" (see [50]). Whilst these responses are explicable, the explicability may, in large part, consist in participants reverting to established display rules for each emotion (i.e., crying when "sad", clenching fists in "anger", and our whole body "lighting up" when feeling "happy"), but without any such essential, universal, biological phenomena. We can, after all, feel sad without the urge to cry, and feel angry without the urge to form a fist; hence, the mapping suggests (although does not determine) a combination of factors: socio-cultural or universally applicable constructs, along with subjective sensations.

This concept-based interpretation may link to some of the other findings, such as the dominance of not only the face, which is understandable given the role of emotional facial displays and startle responses, but also the head. Nummenmaa, Glerean, Hari & Hietanen, [50] suggest that the level of agreement over activation in the head is in part due to "felt changes in the contents of mind triggered by the emotional events" (p. 648). This is, however, a different sense of "sensation" from that which the body maps are intended to display, which is explicitly a matter of capturing actual bodily feelings at a location, such as feeling one's heart increase. It would not seem possible to feel "changes in the contents of mind" in the head, in quite this way; to express so would appear to be drawing on a purely cognitive awareness of mental content coupled with an inference about the brain/mind being located in the head, rather than a sensation experienced in the head. It is not possible to separate out the head from the face in the mapping responses, but if head clicking was even in part due to the reasons stated by the authors, this should provide further reason for caution when interpreting the other responses. However, with indications of "blood rushing to the head" in the traditional self-report studies, supported by physiological measures, to some extent the maps may be drawing on this physical phenomenon, or similar.

The authors' interpretations of the maps produced by children [140,141], people with schizophrenia [146] and older participants [147] point to developmental or organic explanations, with body maps being viewed as essentially biological representations. Again, it may be said that to the extent there are relevant socio-cultural explanations, children will have limited access to the representations of emotion ("net sensations") that are so familiar to adults. Indeed, as "net sensations", these could be interpreted as "generalised instances of emotion-by-situation"; hence, children will not have the experiential base to form such generalisations. This may be one explanation. Another may be more relevant to older participants and those with schizophrenia, who show less differentiation and activation in their maps. Rather than an interoceptive deficit, as suggested in the publications, it may be rather an issue of memory impairment, perhaps more obviously so in the case of older age, although memory impairment is also a symptom of schizophrenia and a side effect of psychoactive medication. Hence, impaired memory may result in reduced access to the situation-indexed array of concepts required to respond in predictable ways on the body maps. Incidentally, one group of authors [147] suggest that the apparently diminished interoceptive ability of older people fits with previous research indicating that older people have greater emotion regulation ability than younger. However, as indicated previously, there is a body of research pointing to *greater* interoceptive ability being vital for greater emotion regulation ability.

Another related issue pertains to the well-known difficulties of engaging in interoception, as indicated in relation to heartbeat detection. The benefits of interoceptive training for interoceptive performance have been shown in multiple studies. For example, Farb, Segal, Mayberg, Bean, McKeon, Fatima & Anderson [36] report an 8-week mindfulness intervention–an investigation of interoceptive sensations–finding that when participants had no mindfulness training there was a coupling of brain regions involved in sensory and conceptual processing. However, after training, these brain regions became decoupled, suggesting that whilst trained individuals can undertake interoception, those without training may struggle to distinguish sensations from sensation concepts [36]. With the body mapping participants seemingly undertaking no such training, there is again reason for caution before drawing definitive conclusions that the maps represent sensations rather than concepts.

A concept-based interpretation is also potentially relevant to the issue of early stage emotional responses, prior to full blown responses. The suggestion has been that embodied emotion research could contribute to the emotion regulation evidence base by identifying rapidly emerging bodily changes as a target for early stage emotion regulation. This regulation could be antecedent-focussed, a strategy in place prior to rapid changes, and therefore able to be effective prior to escalation into full-blown multi-componential syndromes. As indicated previously, in order to research this the timescale for applying a stimulus and recording a response must be brief, but the difficulty with the core texts is that the stimuli were delivered, it seems, over indefinitely long timescales (e.g., [50], or restricted but still many minutes long (e.g., [143,144]), and with no indication of speeded responses having been required. This increases the likelihood that participants could draw on rich appraisals of the stimuli, which, even in those cases of "non-verbal" stimuli, it is possible that the emotional categories to which these belong were clear to participants and, therefore, which appraisals were conceptually appropriate to apply (i.e., which "readout" to draw on) when responding. Indeed, Jung, Ryu, Lee, Wallraven & Chae [139], imply this influence of concepts (p. 10) regarding their use of masking.

Despite the critique thus far, there is a high degree of concordance found within studies that use different languages, from different cultures (see, e.g., [147]), and using multimodal stimuli (e.g., [50]), which retains reason to take seriously the prospect of universal locations of sensation. However, despite applying a relatively low threshold for consistency in a general assessment across the body mapping publications, we still identified a notable lack of agreement regarding which parts of the body change within each "basic emotion", particularly regarding peripheral body parts. Indeed, the only reliably high concordance for activation/deactivation seems to be for the head/face, chest and throat, and with similarly variable consistency when comparing consensuses between emotion types. If the body maps are capturing underlying physiological changes we should perhaps expect greater consistency across studies. Or, perhaps it just is the case (as stated in [147]), that there are other influences on consistency that need to be taken into account. The more consistent body area changes may genuinely be universal, possibly as sensations that "represent a general arousal state resulting from autonomic activation" ([142], p. 6), with the remainder being situationally variable and context-dependent. It is also interesting to note that, in our brief visual assessment, the further away a location on the map was from the head/face, the lower the level of consistency found within and across studies. Again, this could indicate something important about actual bodily activation (e.g., facial muscle movements are a core feature of emotional sensations), or that the maps tap into a concept-based articulation rather than actually mapping bodily sensations (i.e., the possibility that the head/face consistency displays mental "sensations"). The difficulty, then, is how to disentangle genuine sensation reports from concept-based reports without concurrent physiological measurement.

The more traditional self-report studies are only somewhat helpful in this respect, given they do not all provide extensive opportunities for self-report, do not clearly indicate attempts to reduce appraisals of bodily activity, do not report early stage time-stamped self-reports (hence reducing the influence of concepts), and do not objectively measure all self-reported body areas (e.g., the gut). Their value is, however, in supporting the possibility of a few fundamental bodily changes via concordance between self-report and same-location physiological measurement (including the physiological support for head-specific changes that can be sensed as opposed to a change in mental processing). Unfortunately, there is limited ability to make "basic emotion" comparisons with the mapping studies, given that different category names were used ("joy" in [143,144], as opposed to "happiness"; plus "anxiety" as opposed to "fear"), and mixed findings on disgust.

Referring back to the general texts that at least measured all three key visceroceptive domains does little to clarify this situation, with only six texts covering all three, and with a split between subjectivity and objectivity across these. Four of the six [60,106,116,129] collected but did not report data for all three domains, instead combining self-report measurements of body location into composite scores, and otherwise reported physiological measurement of the heart only. The remaining two texts [56,123] illustrated rather the converse situation: recording and reporting only physiological measurements of all three domains. In a study of adult participants, the first of these [56] recorded gastric changes using EGG, respiration with a resistive chest strap, and heart rate by ECG, also measuring RSA. They found no main effects in any domain, with some interactions involving the heart and for RSA but not for respiratory rate. Sternbach [123], however, found changes in all three domains, using an ingested magnet to measure gastric motility changes, a strain gauge around the chest for respiratory changes and heart measurement by ECG, with film clips used as stimuli for child participants. Hence, in the absence of corresponding self-report data, differing measurement tools regarding the gastrointestinal component, child vs. adult samples, bidirectional results, and with one study being over sixty years old, little can be concluded from these studies.

## The future of embodied emotion research

Future research would benefit from, at least, incorporating the three key domains in investigations of embodied emotional feeling, whilst ensuring opportunities for subjective-objective correspondence (but noting difficulties elsewhere in identifying correspondence between interoceptive sensitivity and sensibility). Concurrent physiological measures would be essential in at least these domains, perhaps in combination with body mapping. On this general point, Nummenmaa, Glerean, Hari & Hietanen [50] suggest that future developments in physiological measurement may involve the option of using "whole-body O-H20 PET imaging" (p. 650), which would permit objective assessments matching the self-report responses in body mapping studies. If body maps are used, it would be important to validate these as providing genuine measurements of interoceptive sensibility (or, given, the ongoing development of interoception as a construct, whichever form this will ultimately take). As emphasized throughout, there would also be substantial benefit from focusing on time-limited emotional stimuli presentations, potentially also with the use of masking to thereby limit appraisal. Whilst something like a mask was used in one core text [139], seemingly paradoxically, these authors suggested that their masking prevented "emotional experience", per se, whilst at the same time they sought to address "emotional sensations". In general, to increase the chance of eliciting physiologically detectable bodily changes, validated emotional stimuli should be used (e.g., IAPS). An interoceptive training period could be applied in longitudinal studies, which may help to identify whether improvements in interoception result in greater concordance in

body maps, potentially finding more general patterns of activation and deactivation that cut across "basic emotion" types.

## Limitations

This review is subject to several limitations. Firstly, although four sets of literature searches were undertaken, including an initial set that used a search strategy developed with professional library staff, and across multiple databases, there remains the possibility of bias. The 2014 paper by Nummenmaa, Glerean, Hari & Hietanen [50] was already known to the authors prior to accessing the databases; hence, when this relevant paper was not identified during searches, this gave cause for concern. A related paper [146] was, however, identified using the same initial search strategy, which involved a clinical sample. There was, then, the possibility that the main search strategy tended to identify clinical studies, despite the search terms themselves not restricting in this way, thereby potentially missing a collection of relevant non-clinical studies.

The language restriction decision was based on there being limited resources for accessing and translating all non-English texts that otherwise satisfied the search criteria. This may have resulted in missed relevant texts. Also, in terms of applying criteria, there would ideally have been more than a 10% check of the text selection process; again, due to limitations on resources, this was not possible. Hence, this has increased the chances of error in applying criteria.

Regarding the visual inspection of body maps and subsequent comparisons and contrasts, there may have been room for interpretation regarding classifications of N, T, or perhaps borderline cases for A or D. For the most part, the maps clearly display areas of changed activity, with room for interpretation largely relating to activation in the hands, feet, legs and arms. Due, perhaps, to computer screen limitations, or the quality of online graphics, other visual "traces" of activity on maps may have been missed. However, the images inspected have been indicated in the foregoing text and are available within the cited publications. Also, providing an unweighted simple average of the consistency scores may not be valid, depending on the level of indeterminacy regarding whether one of the two interpretative options has greater 'weight'.

The review has emphasised the tripartite model of interoception developed by Garfinkel and colleagues [5]. This assumes that objective measures of interoception (sensitivity) are possible, and that these can correspond (as metacognitive awareness) to subjective measures of interoception (sensibility)–however, the extent to which this all occurs has been challenged in the literature. Hence, the standard imposed throughout the review regarding the ideal of studies providing objective and subjective measures, a standard in part predicated on the tripartite model, may be considered unnecessarily restrictive. However, this standard is also based on the more general aim of the review: to establish where in the body (an objective matter) emotions are felt (a subjective matter). Hence, even independently of the tripartite view of interoception, at the level of Williams James' original intuitions, subjective-objective correspondence remains a key focus, as opposed to being restricted to a single articulation of one possible construct (interoception) within the broader field of embodied emotion.

## Conclusions

In the attempt to identify empirical studies on where emotional feeling is felt in the body, particularly during the early stages of an unfolding emotion, this review ultimately found no studies meeting a "gold standard". That is, no study undertook concurrent subjective self-report and objective bodily/physiological measurements pertaining to a comprehensive array of body areas including all three key visceroceptive domains; nor were there innovative self-report tools that could clearly approximate the value of physiological measurement.

The few general papers (providing a backdrop to the review's core texts) that did measure all three domains, did so only at the objective level, with varying outcomes. The dominance of heart measurement in the wider literature was confirmed, with perhaps greater dominance than might have been expected. Also confirmed was the widespread scarcity of gut measurement (whether self-report or physiological), which was perhaps more scarce than might have been expected. The gut may be a particularly important area for embodied emotion, given the role of the enteric nervous system, the gut biome and the vagus nerve [19,20].

Taken together, the core texts suggest that the face/head and chest (heart and respiratory activity) may be fundamental in embodied emotion. These were the most consistent self-reports, regardless of emotion type, reports that matched physiological activity in studies that also recorded facial movements (EMG), heart rate (ECG) and respiratory activity (using a thermistor). This does not, however, permit conclusions on early stage sensations, given the absence of early stage subjective reports concurrent with time stamped physiological changes. Further, the current review identified multiple issues with the core texts, which necessarily gives cause for caution when interpreting the self-report findings, in particular, as a matter of emotional sensation. Also, none of the core texts measured physiological changes in gut activity.

Overall, there is significant room for development in this research area, to ground subjective reports of emotional feeling in objective bodily changes, in keeping with the original insights of James [1], and then Damasio [2]. This has the potential to provide greater nuance for understanding location-based contributions to interoception, and for the continued development of clinical intervention tools, building on the evidence base addressing the connections between mindfulness and interoception. The notable lack of studies incorporating gastrointestinal measurement, in particular, despite physiological evidence supporting its role in emotion, should be a focal point in future studies by utilizing existing measurement tools (e.g., electrogastrograph, EGG).

## Supporting information

**S1 Table. Search strategies.**
(DOCX)

**S2 Table. Core text location summaries.**
(DOCX)

**S3 Table. Locations indicated in traditional self-report papers.**
(DOCX)

**S1 Data. Minimal dataset.**
(XLSX)

## Acknowledgments

The authors would like to thank Michael Fauchelle and Susan Hope (Medical and Health Sciences Library, University of Otago Wellington) for their guidance in the development of the search strategy for this review.

## Author Contributions

**Conceptualization:** Steven Davey.

**Data curation:** Steven Davey.

**Investigation:** Steven Davey.

**Methodology:** Steven Davey.

**Project administration:** Steven Davey.

**Supervision:** Jamin Halberstadt, Elliot Bell.

**Writing – original draft:** Steven Davey.

**Writing – review & editing:** Steven Davey, Jamin Halberstadt, Elliot Bell.

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
