## [Decision Letter · Decision Letter 0]

19 Nov 2021

PONE-D-21-12359Where is emotional feeling felt in the body? An integrative reviewPLOS ONE

Dear Dr. Davey,

Thank you for submitting your manuscript to PLOS ONE. After careful consideration, we feel that it has merit but does not fully meet PLOS ONE’s publication criteria as it currently stands. Therefore, we invite you to submit a revised version of the manuscript that addresses the points raised during the review process.

We look forward to receiving your revised manuscript.

Kind regards,

Zezhi Li, Ph.D., M.D.

Academic Editor

PLOS ONE

Journal Requirements:

2. Please include your tables as part of your main manuscript and remove the individual files. Please note that supplementary tables (should remain/ be uploaded) as separate "supporting information" files

Reviewers' comments:

Reviewer's Responses to Questions

**Comments to the Author**

1. Is the manuscript technically sound, and do the data support the conclusions?

Reviewer #1: Yes

2. Has the statistical analysis been performed appropriately and rigorously? 

Reviewer #1: N/A

3. Have the authors made all data underlying the findings in their manuscript fully available?

Reviewer #1: Yes

4. Is the manuscript presented in an intelligible fashion and written in standard English?

Reviewer #1: Yes

5. Review Comments to the Author

Reviewer #1: I think this article merits publication as it addresses gaps in the literature. Trying to link the physiological responses with the anatomical substrates to the subjective emotional dimension under the sense of interoception is insightful and, I believe, should move the field forward. The methodology and rationale were sound. The article was clear and well-written.

The issues that need to be addressed were the assumptions and models used to explain interoception and the “visceroceptive” processes. For instance, the Garfinkel et al., (2015) model is not without critique. Both Mehling (2016) and Gibson (2019) address some of the limitations of said model. What current research is showing is that the objective dimensions of interoception have little bearing on the subjective dimensions – which seemed to bear out in the authors’ review. There are a number of ways the authors can address this specific issue. One is to highlight the contributions as well as the limitations of using Garfinkel et al., model. Another is to introduce other models that explore the multidimensionality of interoception (i.e., Mehling et al., 2012; Gibson, 2019). Given that the authors using the Garfinkel model which is the lynchpin to the entire article, these limitations need to be addressed. As Mehling (2016) and Ferentzi et al., (2018, 2019) point out, the complexity of interoception is difficult to measure. The objective, physiological dimensions don’t map onto subjective measures easily, if at all, nor may they have any impact on the subjective dimensions of interoception.

As for the visceroceptive processes, there ought to be justification for the focus here. Much of the interoception literature focuses on the internal state of the body, not only or specifically the visceroceptive processes. For instance, cardioception (usually measured by the heart-beat detection task) has come under a lot of scrutiny lately (Ferentzi and others). The HBD detection task was used because it was easy to measure and assumed to reflect a general sensitivity for other visceral processes and interoception broadly. Recent research has not confirmed these assumptions. These limitations need to be addressed in this article – not just in the introduction, but throughout including the results and conclusion sections.

One final note, I do think the authors can expand a little more on Damasio’s somatic marker hypothesis in the introduction as it would only serve to buttress the core of their argument. All in all, this article can be a valuable, contributing piece in the literature.

6. PLOS authors have the option to publish the peer review history of their article (what does this mean?). If published, this will include your full peer review and any attached files.

Reviewer #1: No

---

## [Author Response · Author response to Decision Letter 0]

29 Nov 2021

For the attention of the Editor and Reviewer:

We are hereby submitting a revised version of a review paper with the title, “Where is emotional feeling felt in the body? An integrative review”. We have responded to each point raised by the Editor and the Reviewer in the manuscript and provide a summary here of our responses.

Firstly, we have altered the format of the paper to bring this in line with PLOS ONE’s style requirements, including changes to the reference style (from APA to Vancouver).

Secondly, all tables are now part of the main manuscript, and the figures have been entered into PACE as instructed, with the adjusted versions uploaded. 

Thirdly, we have provided a “minimal data set” as a Supporting Information file. This lists all information extracted from the papers that provide the basis for the review, along with PRISMA results breakdown, and the raw data underlying the main graph presented in the first part of the review. 

Fourthly, in relation to the Reviewer’s comments, we would first like to thank them for their instructive, helpful comments, and for their general support of the paper. In relation to their specific comments, we take each in turn:

1. “Given that the authors using the Garfinkel model which is the lynchpin to the entire article, these limitations need to be addressed. As Mehling (2016) and Ferentzi et al., (2018, 2019) point out, the complexity of interoception is difficult to measure. The objective, physiological dimensions don’t map onto subjective measures easily, if at all, nor may they have any impact on the subjective dimensions of interoception.”

a. In response: we have now provided criticisms of the Garfinkel model (pp. 6-7, 10-11, 57, and 66-67 of the main document). We have also made further references to the MAIA (the model provided by Mehling and colleagues), as suggested by the Reviewer. 

b. We have made explicit a yet more fundamental ‘lynchpin’: the review’s focus on embodied emotion, principally the work of William James and Damasio. Whilst interoception is a prominent aspect of the paper, the intention of including this was not to focus solely on this construct, but to emphasise it as a current, arguably dominant articulation of the general idea of embodied emotion. The essence of the latter is that the objectivity of the body informs the subjectivity of [emotional] experience. Interoception is, in our view, currently the most encouraging specific articulation of the general concept of embodiment. Further, the intention of focusing on the Garfinkel model was, again, that it is (arguably) the most prominent current articulation of interoception. Hence, at various points in the paper (pp. 4, 9, 10-11, 56, 66-68), we have brought this point out more clearly.

2. “As for the visceroceptive processes, there ought to be justification for the focus here. Much of the interoception literature focuses on the internal state of the body, not only or specifically the visceroceptive processes. For instance, cardioception (usually measured by the heart-beat detection task) has come under a lot of scrutiny lately (Ferentzi and others). The HBD detection task was used because it was easy to measure and assumed to reflect a general sensitivity for other visceral processes and interoception broadly. Recent research has not confirmed these assumptions. These limitations need to be addressed in this article – not just in the introduction, but throughout including the results and conclusion sections.”

a. One of our responses was to provide further background (pp. 4-5) regarding the relevance of the viscera to interoception, but particularly in their relationship with emotion. This subset of the interoception research literature has emphasised the viscera (we cite two prominent review papers, one by Critchley and Garfinkel, and one by Cameron, both of which heavily emphasise the viscera; these pick up on a wide literature, encompassing also clinical issues pertaining to the viscera, including panic disorder and irritable bowel syndrome). Rightly or wrongly, this is the direction the field has taken; thereby, we picked up on this in the review. That said, we do not limit ourselves to only these three visceral systems during the process of article selection or when assessing the articles. Indeed, ideally, research studies should gather data on all systems of the body. But, we recognised that this would be unrealistic, even for our “gold standard”; hence, we stipulated that a “gold standard” article is one where “at least a majority of overall body locations implicated in the literature (including all three visceroceptive systems) are measured”. In this way, we aimed to access studies that had made a serious attempt to answer the question of our review of “where emotions are felt”, which thereby included those interoceptive studies that had tended to emphasise the viscera.

b. Another response was to point out what the Reviewer suggested regarding the attraction of the Schandry task (i.e., it is easy to administer; see p. 5), which may in part explain why it is so often the focus, alongside criticisms of the task (p. 10). Further, again as the Reviewer pointed out, the lack of correspondence across interoceptive channels (i.e., being good at the Schandry task need not generalise across non-cardiac channels) has now been included (pp. 6-7, 57) and in fact used to bolster part of the rationale for the review (i.e., to challenge cardiac dominance: it does not make sense to limit the focus in this way when we cannot generalise from one channel to any/all other channels; p. 10).

3. “One final note, I do think the authors can expand a little more on Damasio’s somatic marker hypothesis in the introduction as it would only serve to buttress the core of their argument. All in all, this article can be a valuable, contributing piece in the literature.”

a. We have now added in more detail on Damasio (pp. 3-4, 11).

We hope these revisions have done justice to the Reviewer’s input, and look forward to hearing the outcome. 

Regards,

Dr. Steven Davey (on behalf of co-authors Prof. Jamin Halberstadt and Dr. Elliot Bell).

---

## [Decision Letter · Decision Letter 1]

9 Dec 2021

Where is emotional feeling felt in the body? An integrative review

PONE-D-21-12359R1

Dear Dr. Davey,

We’re pleased to inform you that your manuscript has been judged scientifically suitable for publication and will be formally accepted for publication once it meets all outstanding technical requirements.

Kind regards,

Zezhi Li, Ph.D., M.D.

Academic Editor

PLOS ONE

Additional Editor Comments (optional):

Reviewers' comments:

Reviewer's Responses to Questions

**Comments to the Author**

1. If the authors have adequately addressed your comments raised in a previous round of review and you feel that this manuscript is now acceptable for publication, you may indicate that here to bypass the “Comments to the Author” section, enter your conflict of interest statement in the “Confidential to Editor” section, and submit your "Accept" recommendation.

Reviewer #1: All comments have been addressed

2. Is the manuscript technically sound, and do the data support the conclusions?

Reviewer #1: Yes

3. Has the statistical analysis been performed appropriately and rigorously? 

Reviewer #1: I Don't Know

4. Have the authors made all data underlying the findings in their manuscript fully available?

Reviewer #1: Yes

5. Is the manuscript presented in an intelligible fashion and written in standard English?

Reviewer #1: Yes

6. Review Comments to the Author

Reviewer #1: This version has a better theoretical grounding and sound rationale which supports the primary purpose of the paper. The authors did a nice job of responding to prior comments and adequately addressed the concerns that were raised. I have no qualms recommending this piece for publication.

There is one comment I would make in regards to the Aims section on page 12. The authors highlight the challenges of what to measure and how and the lack of correspondence between sensitivity and sensibility. A footnote can be made that top-down processes (i.e., memory, expectations) can alter interoceptive sensibility - which this concepts extends beyond the scope of this paper nor does it address the two main objectives - thus the footnote. However, this comment can be made acknowledge these methodological challenges in the interoceptive domain as well. It might also partly explain why sensitivity and sensibility do not always overlap.

7. PLOS authors have the option to publish the peer review history of their article (what does this mean?). If published, this will include your full peer review and any attached files.

Reviewer #1: No

---

## [Editor Report · Acceptance letter]

13 Dec 2021

PONE-D-21-12359R1 

Where is emotional feeling felt in the body? An integrative review 

Dear Dr. Davey:

I'm pleased to inform you that your manuscript has been deemed suitable for publication in PLOS ONE. Congratulations! Your manuscript is now with our production department. 

Kind regards, 

on behalf of

Dr. Zezhi Li 

Academic Editor

PLOS ONE